# Humans need auditory experience to produce typical volitional nonverbal vocalizations
Katarzyna Pisanski [1,2,3] ✉, David Reby [1,4,6] & Anna Oleszkiewicz [5,3,6] ✉

Human nonverbal vocalizations such as screams and cries often reflect their evolved functions. Although the universality of these putatively primordial vocal signals and their phylogenetic roots in animal calls suggest a strong reflexive foundation, many of the emotional vocalizations that we humans produce are under our voluntary control. This suggests that, like speech, volitional vocalizations may require auditory input to develop typically. Here, we acoustically analyzed hundreds of volitional vocalizations produced by profoundly deaf adults and typically-hearing controls. We show that deaf adults produce unconventional and homogenous vocalizations of aggression and pain that are unusually high-pitched, unarticulated, and with extremely few harsh-sounding nonlinear phenomena compared to controls. In contrast, fear vocalizations of deaf adults are relatively acoustically typical. In four lab experiments involving a range of perception tasks with 444 participants, listeners were less accurate in identifying the intended emotions of vocalizations produced by deaf vocalizers than by controls, perceived their vocalizations as less authentic, and reliably detected deafness. Vocalizations of congenitally deaf adults with zero auditory experience were most atypical, suggesting additive effects of auditory deprivation. Vocal learning in humans may thus be required not only for speech, but also to acquire the full repertoire of volitional non-linguistic vocalizations.

Nonverbal vocalizations occupy an evolutionarily primitive niche in the human vocal repertoire[1,2]. These non-linguistic vocal sounds such as cries, laughter, and screams, communicate emotions and motivations, and almost certainly emerged before words in our ancestral lineage[3,4]. The acoustic forms of human nonverbal vocalizations often reflect their evolved functions (ref. 2 for review) and appear largely homologous to the evolved affective vocal displays of other animals, including our closest primate relatives[5–8]. Growing evidence that human vocalizations share key similarities across cultures[2,9,10] has also led researchers to debate the possibility that some nonverbal vocalizations may represent universal communicative signals[2,11]. But how do we humans acquire them?

Unlike spoken language that must be acquired through vocal production learning[12,13], some forms of spontaneous nonverbal vocalizations are innate in humans, suggesting they can emerge without any sensory input or vocal motor experience. For example, human babies across the globe are born crying. Already at birth, the harsh and chaotic sounds of their cries function to exploit the perceptual sensitivities of their caregivers because such cries are both aversive and extremely difficult to ignore[14,15]. The universal function of the human baby cry is thus to elicit aid, not unlike the distress calls of other mammalian infants, and its acoustic form is well suited to this function[2]. Indeed babies' cries are an excellent example of form-function mapping in human vocalizations[2]. However, as humans age, a broader array of vocalization types characterizes interpersonal interactions, from laughter that can signal both affiliation and malice[16,17] to aggressive roar-like calls that can signal strength or intimidation[18–20]. Although these vocalization types emerge later in human ontogeny than crying does, their stereotyped acoustic forms could still be guided by specialized innate mechanisms (i.e., biological preparedness) or species-specific templates[21], and these mechanisms and templates may require auditory input throughout the lifespan to mature typically. Critically, in humans, many of these emotional vocalizations are also produced volitionally in adulthood[2], making them prime suspects for vocal production learning, as discussed below.

[1]ENES Bioacoustics Research Laboratory, CRNL Center for Research in Neuroscience in Lyon, University of Saint-Étienne, 42023 Saint-Étienne, France. [2]CNRS French National Centre for Scientific Research, DDL Dynamics of Language Lab, University of Lyon 2, 69007 Lyon, France. [3]Institute of Psychology, University of Wrocław, 50-527 Wrocław, Poland. [4]Institut Universitaire de France, Paris, France. [5]Department of Otorhinolaryngology, Smell and Taste Clinic, Carl Gustav Carus Medical School, Technische Universitaet Dresden, 01307 Dresden, Germany. [6]These authors contributed equally: David Reby, Anna Oleszkiewicz. ✉e-mail: katarzyna.pisanski@cnrs.fr; ania.oleszkiewicz@gmail.com

When we refer to the typical maturation of vocalizations, we are referring to the development of nonverbal vocalizations that have stereotyped and predictable acoustic forms depending on their intended emotion or motivation (i.e., form-function mapping). In control vocalizers with fully intact hearing, the acoustic forms of vocalizations often reflect the social functions of those calls in ways that align with the motivational-structural rules that also characterize other animal calls[2]. For example, fear screams in humans and other animals are typically extremely high-pitched as they function to communicate alarm[22–24], whereas aggressive roars are often much lower pitched as they function to intimidate[18,20]. Both call types are thus "designed" to exploit the hearing sensitivities and perceptual biases of listeners. Emotional vocalizations such as these also often sound acoustically harsh. This harshness arises from nonlinear phenomena caused by aperiodic vibration of the vocal folds that make the calls perceptually rough and unpredictable, and thus ostensibly difficult to ignore and habituate to[25–27]. Nonlinear phenomena (NLP) remain understudied in the human voice sciences compared to the fundamental frequency ($f_o$, perceived as voice pitch) and formant frequencies (resonances of the vocal tract), both of which have been instrumental in understanding the mechanisms and evolved functions of animal calls and human speech[28]. Nevertheless, recent work shows that nonlinear phenomena are differentially processed in the human brain relative to tonal sounds[27] and may have important social functions, such as communicating distress[15,29], arousal[30] or threat and formidability[18–20,26] in human nonverbal vocalizations, not unlike their apparent functions in other animal calls[25].

While the forms and functions of many human vocalizations appear homologous to those of other animals, and thus share many similarities, there is one critical difference: humans can voluntarily produce vocal sounds more effortlessly than any other primate species[3,31]. Although other primates, including great apes, do show some degree of vocal flexibility (refs. 32,33 for reviews), their ability to control their vocal output is extremely rudimental compared to that of humans[3]. Humans can easily exaggerate vocalizations voluntarily (e.g., an embellished pleasure moan), produce completely 'fake' vocalizations (such as when we laugh at a joke that is not at all funny), or produce sounds on demand in the absence of any external or internal trigger. These volitional vocalizations in humans readily implicate higher cortical pathways shared with speech production, whereas more reflexive and less intentional call production activates the ancient subcortical limbic system, including the anterior cingulate cortex in humans and other mammals alike[3,4,21,31]. Because human volitional vocalizations appear to tap into similar neural pathways as speech does, they may, like speech, require auditory feedback to mature in a typical or conventional manner. This is the key hypothesis that we set out to test in this paper.

One way to test this hypothesis is by examining volitional vocalizations in persons with limited to no auditory experience. Studying vocal communication in people with hearing impairments can help to create interventions, where needed and desired, that can facilitate interpersonal communication. It can also provide unique empirical insights into how external auditory input (acoustic signals processed by specialized sensory organs) or internal auditory feedback (hearing oneself vocalize) shape communicative signals. In nonhuman animals, the behaviors of deafened individuals have traditionally been contrasted with those of individuals placed in social isolation to determine the extent to which vocalization types are shaped by sensory input[34,35]. Such experiments have largely focused on songbirds as they are arguably the greatest vocal production learners of the animal kingdom, alongside humans. These studies show that songbirds often develop abnormal songs if deafened during early development or if socially isolated and prevented from hearing the songs of conspecifics. In contrast, if socially isolated but exposed to song recordings, many songbirds develop relatively natural songs. Thus in these vocal learning species, auditory experience is necessary and sufficient for typical vocal development[34].

The role of auditory input in the vocal production of nonhuman mammals is lesser studied and remains debated[35,36]. Some mammals, including cats[37] and bats[38] produce atypical vocalizations when deafened early in life, while others show evidence of innate motor programs, such as

mice that appear to produce species-typical ultra-sonic courtship vocalizations into adulthood in the absence of any auditory input[39] (but see[34]). Although deafening and isolation experiments in primates are very rare, the vocal repertoires of squirrel monkeys also remain largely intact despite early-deafening or isolation[40]. This is consistent with a general lack of evidence for vocal production learning in nonhuman primates, including great apes[36,41,42]. Notably, however, there is mounting evidence for vocal plasticity and a developmental role of experience in shaping the vocalizations of some nonhuman primates, such as marmosets[43,44], as they age.

In humans, it is well established that speech development requires auditory feedback and vocal production learning at a critical period before adolescence[12]. Without it, congenitally deaf adults often show profound speech impairments. Individuals deafened post-lingually also produce atypical vowel, consonant, and suprasegmental speech sounds[45,46], indicating that auditory feedback regulates speech production even after language is acquired, and possibly in real-time. Yet few studies have examined how auditory deprivation affects *non-linguistic* vocal signals in humans, and the few studies that exist have focused almost exclusively on cries and laughter. For instance, early case studies suggested that deaf infants produce normative cries at birth, but showed some evidence that cries may be marked by progressive acoustic atypicality as infants age[47–51]. This suggests that continuous auditory feedback may be needed to retain typical cry structure. In contrast, a small number of studies on adults with hearing impairments showed that the acoustic structure of spontaneous laughter is broadly typical in form, albeit with some differences in fundamental frequency, duration, amplitude, and voicing, compared to the laughter of adults with typical hearing[52]. Another study found that laughter is used to punctuate conversation in sign language, much as it does in spoken language, pointing to a conserved pragmatic communicative function across human language systems[53].

Although these few studies suggest that spontaneous, reflexive laughter may require little auditory input to mature relatively typically, this may not be the case for laughter that is produced volitionally. Voluntary vocal expressions might require more extensive vocal learning or vocal motor experience than spontaneous ones. This is because the underlying brain mechanisms that drive spontaneous emotional vocalizations tap into deep, primitive regions of the brain shared by all mammals, whereas volitional vocalizations tap into higher-order cortical pathways shared with speech production, i.e., the central or core "speech centers" of the brain[31]. We thus predict that auditory deprivation is likely to lead to atypicality, particularly in *volitional* vocalizations, more so than in spontaneous ones, much as it does for speech. Research on blind adults supports this prediction in the context of visual deprivation. Blindness moderates the production of facial expressions of volitional (posed or acted) emotions, but not of spontaneous facial expressions produced in response to real-life emotional experiences, such as triumph or defeat in the Olympic games (ref. 54 for review).

It has yet to be tested in humans whether auditory deprivation will more severely impair volition than spontaneous vocalizations. In one related study, however, Sauter and colleagues[55] examined the production and perception of negative and positive vocalizations produced solely on demand by eight hearing-impaired adults. Half of these eight adults had used a hearing aid, leaving the question open about how complete sensory deprivation affects volitional vocalizations in humans. Although the authors argued that the vocal sounds of hearing-impaired adults were broadly typical in acoustic form, they also found structural abnormalities in virtually every measured voice parameter. Moreover, although listeners could judge the intended emotions of most call types above chance levels, they performed substantially worse when discriminating the volitional vocalizations of deaf vocalizers than those of healthy hearing controls and were not able to correctly identify anger in deaf vocalizers. It thus remains unclear how exactly auditory deprivation influences voluntary vocalizations in humans and, critically, whether the effects of auditory deprivation are additive such that less auditory experience leads to more vocal atypicality.

Here, in a large sample of sixty profoundly hearing-impaired adults with different measurable degrees of hearing loss, and sixty typically hearing

controls matched for sex and age with our sample of deaf adults, we address several long-standing questions. How do humans acquire their extensive nonverbal vocal repertoires? Do volitional vocalizations such as cries, roars, and screams require auditory input to cultivate their stereotyped forms into adulthood? Does the extent of a vocalizer's auditory experience (or lack thereof) throughout the lifespan predict how typical their vocalizations sound, and how reliably those vocalizations communicate their intended emotions to listeners?

To this aim, each vocalizer in our study intentionally produced calls of aggression, pain, and fear. Using acoustic analysis, we tested the prediction that the nonverbal vocalizations of deaf adults would differ structurally from those of hearing controls, deviating from the acoustic forms we would expect based on the intended motivation or emotion of each call type. In other words, we expected weaker form-function mappings in the vocalizations of deaf adults compared to controls. For example, aggressive calls are typically harsh, low-pitched, loud, and produced with closely spaced formants in typically hearing adults[2], and we predicted these acoustic markers of aggression would be weaker or absent in the vocalizations of deaf vocalizers. Our acoustic models included fifteen perceptually and socially relevant vocal parameters[2,56], such as fundamental frequency (pitch), formant spacing, amplitude, duration, perturbation parameters, and nonlinear phenomena (harshness). While all sixty hearing-impaired volunteers had a clinical diagnosis of bilateral profound deafness as confirmed by our hearing tests, twenty were congenitally deaf and had never used a hearing aid nor cochlear implant, and thus had never heard any external sound in their lifetimes. We directly tested whether these adults with complete auditory deprivation showed the greatest vocal atypicality.

In a series of perception experiments in the lab with 444 human listeners, we then tested how the vocalizations of deaf adults were perceived compared to those of controls. First, we predicted that listeners would be less accurate in correctly recognizing the intended emotions and valences of vocalizations produced by deaf compared to typically hearing adults. We used both a forced-choice paradigm to precisely measure accuracy and confusion rates in emotion judgments (experiment 1), and an open-ended response paradigm to explore broader patterns in listeners' perceptions of emotion and valence without imposing labels (experiment 2). In experiment 3, we tested the prediction that listeners would judge the vocalizations of deaf individuals as less authentic than those of hearing controls. Emotion authenticity ratings can index how convincingly a given vocalization conveys an emotion[57]. Generally, the stronger the acoustic form-function mapping, the more authentic a vocalization tends to sound[58], suggesting that acoustically atypical expressions of aggression, fear, and pain will be judged as relatively inauthentic. Finally, in experiment 4, we tested the prediction that listeners would be able to detect which vocalizations were produced by deaf versus hearing vocalizers, even in the absence of linguistic cues. We expected that listeners would be sensitive to deviations in typical nonverbal vocal patterns in vocalizations, as they are in speech. Taken together, we combine acoustic analyses with perception experiments to provide converging evidence that auditory deprivation impedes not only the encoding of socially relevant information in human vocalizations, but also its decoding by listeners.

## Methods
This research combines several quantitative methodologies, including voice recording, acoustic analysis, and a series of four independent perception experiments which were not preregistered. All research protocols were approved by the Institutional Review Board at the University of Wrocław (IPE0021) in consultation with the Polish Association of the Deaf. The research was performed in accordance with the Declaration of Helsinki for research with human subjects. Informed and written consent was obtained from all vocalizers (n = 120) and listeners (n = 444) before taking part in the research.

### Vocalizers
One hundred and twenty adults representing a broad age range and a balanced sex ratio were audio recorded while producing nonverbal vocalizations in individual private sessions. The sample included 60 profoundly deaf adults (30 male vocalizers, mean age ± sd 29 ± 11.7, age range 16–53 years; 30 female vocalizers, 30 ± 11.6, 17–52 years old) and 60 typically hearing controls matched by sex, age, and education level to the deaf sample (30 male vocalizers, 30 ± 10.6, 16–55 years old; 30 female vocalizers, 30 ± 11.2, 19–55 years old). Vocalizer sex was self-reported and data on ethnicity were not collected. Table 1 provides detailed descriptive statistics for our vocalizer samples, including onset, duration, and causes of deafness.

All deaf participants qualified as profoundly bilaterally deaf, but the severity of deafness varied across vocalizers. Nearly 70% (n = 41) were congenitally deaf. The remainder lost their hearing in infancy or childhood (mean age of onset 3 ± 3.2 years old). As described below and in Table 1, twenty-five individuals had no hearing support, twenty used a hearing aid, and fifteen had cochlear implants at the time of the study. We identified individuals with zero auditory experience as those who were born deaf and never used a cochlear implant or hearing aid (n = 20, see Table 1). Deaf participants were recruited via advertisements and professional contacts with local associations or specialized schools for deaf persons. Age-sex-matched hearing controls were recruited from the general population and local community through personal and professional contacts. To take part in the study, controls must have declared normal hearing, whereas deaf participants must have received a clinical diagnosis of bilateral profound deafness, which we confirmed prior to their taking part in the study.

### Hearing impairment and auditory screening
All 120 vocalizers completed two established screening procedures for speech intelligibility to verify their hearing status: a vocal audiometry matching task in which participants were asked to identify monosyllabic target words from a list of phonetically comparable words presented at variable intensity levels[59,60], and a speech-in-noise digit triplet task in which participants were asked to identify three single-digit numbers embedded in continuous white noise at variable single-to-noise ratios[60,61]. Auditory screening tests were conducted in the lab using a web-based computer platform calibrated with a model-specific, biological sound level reference with a hearing threshold verified with pure-tone audiometry[62,63]. Deaf participants using a hearing aid (n = 20, Table 1) completed the tests with the hearing aid intact for a maximally conservative measure of hearing impairment. Auditory stimuli were played through Sennheiser HD-280 professional headphones, and each ear was tested separately, producing 50% and 100% intelligibility thresholds for both the left and right ear [see ref. 60 for additional details regarding screening procedures].

Auditory screening results are summarized in Table 1. All hearing participants reached 100% intelligibility thresholds in both auditory tests in at least one ear. Nearly 90% of deaf participants scored 0% intelligibility on both tests in both ears, indicating absolutely no ability to comprehend speech sounds, including, of course, the 20 deaf participants with zero hearing experience. The remaining eight deaf participants reached 50% intelligibility in speech audiometry at a sound pressure level (SPL) ranging from 18 dB to 47 dB, and 100% intelligibility at a sound pressure level ranging from 30 dB to 80 dB for the right ear. These eight subjects could thus comprehend 50% of speech sounds when the sound pressure level ranged from 22 dB to 50 dB and 100% at 70 dB with their left ear. One deaf subject reached 50% intelligibility in the digit triplets test at a sound-to-noise ratio (SNR) of −10.25 with the left ear but did not reach 100% intelligibility. All deaf participants declared a minimum 90 dB hearing threshold.

### Voice recording
All vocalizers (n = 120) were recorded privately in a quiet room using a Tascam DR05 recorder at a sampling rate of 48 kHz and 24-bit amplitude quantification, positioned 150 cm from the mouth to avoid audio clipping of high-amplitude vocalizations[18]. Microphone distance and input levels were standardized between and within vocalizers. Instructions were given to all participants in written form before voice recording, and for deaf participants were also provided in sign language via a pre-recorded video featuring a Professional Sign Language Interpreter, who was also available in person on

**Table 1 | Sample characteristics of 60 typically hearing vocalizers (controls) and 60 deaf vocalizers**

| | | Typically hearing vocalizers | | Deaf vocalizers | | | |
|---|---|---|---|---|---|---|---|
| | Statistic | Total | | Total | Hearing aid | Cochlear implant | No hearing support |
| Sex | N | 60 | | 60 | 20 | 15 | 25 |
| | Male vocalizers | 30 | | 30 | 7 | 7 | 16 |
| | Female vocalizers | 30 | | 30 | 13 | 8 | 9 |
| Age | M years | 29.9 | | 29.5 | 27 | 18.7 | 37.9 |
| | SD | 10.8 | | 11.5 | 9.7 | 2.2 | 9.9 |
| | Min | 16 | | 16 | 16 | 16 | 19 |
| | Max | 55 | | 53 | 46 | 23 | 53 |
| Duration of deafness | M years | | | 28.7 | 25.8 | 17.2 | 38.7 |
| | M % of life | | | 95% | 94% | 92% | 99% |
| | SD | | | 12.3 | 10.4 | 4 | 9.2 |
| | Min | | | 9 | 12 | 9 | 19 |
| | Max | | | 53 | 46 | 23 | 53 |
| Congenital deafness | N | | | 41 | 10 | 11 | 20[3] |
| Marginal hearing of sounds | N | 60 | | 35 | 12 | 13 | 10 |
| Age of deafness onset [years] | M | | | 3 | 2.3 | 5.6 | 2.1 |
| | SD | | | 3.2 | 2.6 | 4.6 | 1.7 |
| | Min | | | 0.4 | 0.5 | 0.4 | 0.66 |
| | Max | | | 10 | 8 | 10 | 4 |
| Vocal audiometry n subjects [at min dB][1] | 50% right | 59 [4] | | 6 [18] | 0 [-] | 6 [18] | 0 [-] |
| | 100% right | 59 [10] | | 3 [30] | 0 [-] | 3 [30] | 0 [-] |
| | 50% left | 58 [2] | | 4 [22] | 0 [-] | 4 [22] | 0 [-] |
| | 100% left | 58 [10] | | 1 [70] | 0 [-] | 1 [70] | 0 [-] |
| Digit triplet test n subjects [at min SNR][2] | 50% right | 59 [−20.7] | | 1 [−.8.5] | 0 [-] | 1 [−.8.5] | 0 [-] |
| | 100% right | 59 [−18] | | 0 [-] | 0 [-] | 0 [-] | 0 [-] |
| | 50% left | 59 [−20.4] | | 1 [−10.25] | 0 [-] | 1 [−10.25] | 0 [-] |
| | 100% left | 59 [−16] | | 0 [-] | 0 [-] | 0 [-] | 0 [-] |
| Cause of deafness | Idiopathic [n] | | | 30 | 3 | 9 | 18 |
| | Genetic [n] | | | 7 | 2 | 2 | 3 |
| | Post-operation/disease complications [n] | | | 22 | 15 | 3 | 4 |
| | Post-traumatic [n] | | | 1 | 0 | 1 | 0 |

1 Number of vocalizers who reached 50/100% intelligibility in right/left ear [at min dB HL, decibels hearing level].
2 Number of vocalizers who reached 50/100% intelligibility in right/left ear [at min SNR, signal-to-noise ratio].
3 Congenitally deaf individuals with no cochlear implant and no history of using a hearing aid and thus zero auditory experience.

site if additional communication or translation was required. Vocalizers were instructed to imagine themselves in three scripted scenarios presented in a random order (representing contexts of aggression, fear, and pain). Vignettes were based on prior research on volitional human vocalizations[18,29]. Vocalizers were then asked to respond vocally but non-linguistically (without the use of words) to each given scenario, producing one vocal output per scenario. During the voice recording task, the researcher and sign language interpreter briefly left the room to ensure that the vocalizer felt maximally comfortable producing the vocalizations. This protocol resulted in three vocal stimuli per vocalizer and a total of 360 vocal stimuli. For complete instructions and scenario scripts, see Supplementary Methods. Voice recordings are freely available for research purposes.

## Acoustic analysis

Acoustic analyses were performed using open-source acoustic analysis software *Praat* v 6.1.21[64] and the *R* package *soundgen*[65]. Acoustic measures were taken from the entire vocal stimulus produced by each vocalizer in each context, which sometimes included several voiced segments. We first measured 14 nonverbal acoustic parameters (Supplementary Tables 1, 2) with a custom *Praat* script and an established verification process[29,66]. These

included fundamental frequency parameters (mean $f_o$, min and max $f_o$, $f_o$CV as the coefficient of variation of the pitch contour) measured with a search range of 60–2000 Hz, 0.05 s window length and 0.01 timestep. Smoothing algorithms were then applied to the pitch contour using either a broad or narrow bandwidth to measure major $f_o$ modulations (inflex2) and minor vibrato-like inflections (inflex25), respectively. Extracted $f_o$ contours were systematically inspected and verified for accuracy, and any measurement errors (e.g. octave jumps or tracking errors owing to nonlinear phenomena) were de-selected or corrected. Mean $f_o$ was measured in Hertz and additionally transformed into equivalent rectangular bandwidth units (ERBs, where $Ei = 21.4* \log10(0.00437*fi + 1)$, a quasi-logarithmic scale that accounts for the nonlinear relationship between fundamental frequency and perceived pitch[67]. However, as the Hz and ERB measures were highly correlated ($r = 0.97$) we report results solely in Hz.

Amplitude parameters included mean intensity (mean AMP), max intensity (maxAMP), and intensity variability (intCV, the coefficient of variation of the intensity contour). Noise was measured as harmonics-to-noise ratio (HNR), and frequency and amplitude perturbation were measured as jitter and shimmer, respectively. Finally, the script computed two temporal parameters: the duration of the entire vocal stimulus (dur,

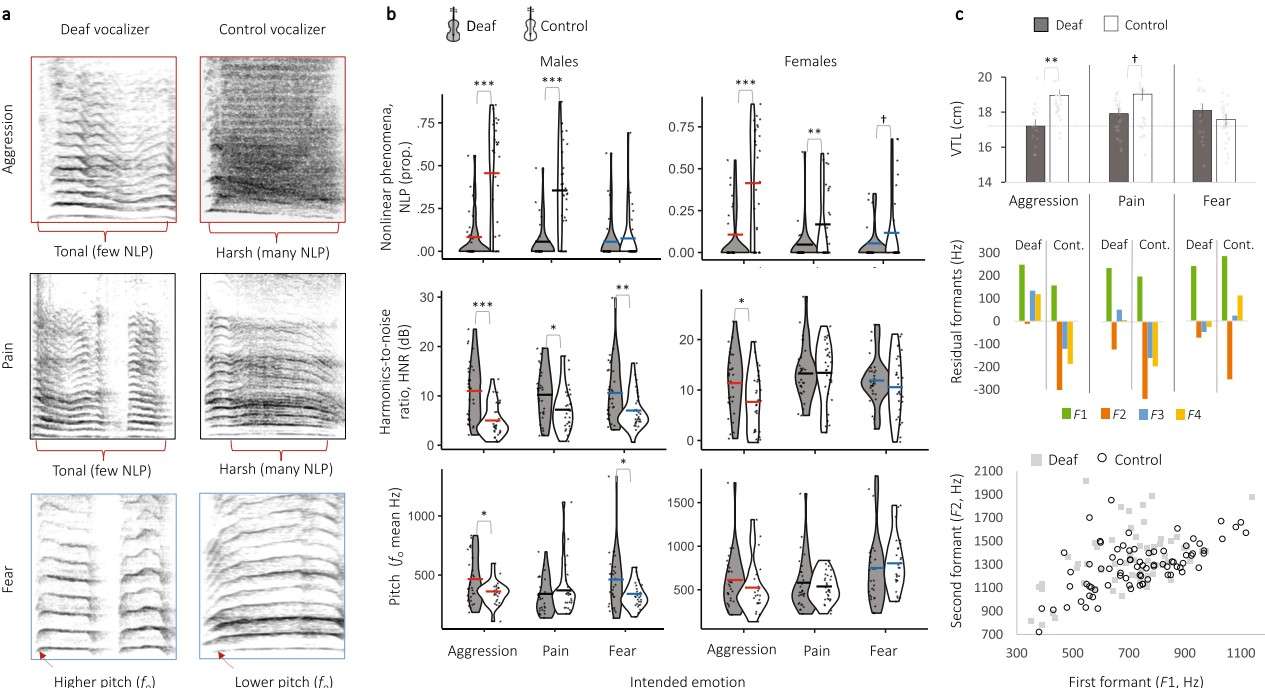

**Fig. 1 | Deaf adults produced acoustically atypical vocalizations that were much more tonal, higher pitched, and less articulated compared to hearing controls.**
**a** Example spectrograms illustrate that vocalizations produced by deaf adults ($n = 60$) were relatively more tonal, less harsh, and higher pitched than were those of typically hearing controls ($n = 60$), among other acoustic differences. Nonlinear phenomena (NLP) were notably several times more common in the aggressive and pain vocalizations of control vocalizers than of deaf vocalizers. **b** Violin plots and overlaid dot plots (datapoints are intentionally jittered for improved visualization) show raw distributions of three key acoustic parameters that were atypical in the vocalizations of adults with deafness ($n = 60$ vocalizers, 180 vocalizations) compared to typically hearing controls ($n = 60$ vocalizers, 180 vocalizations): fewer NLP (proportion of nonlinear phenomena), higher HNR (harmonics-to-noise ratios) and higher pitch (measured as $f_o$). Solid dashes on each violin plot indicate estimated marginal means from LMMs (Supplementary Tables 1, 2). Asterisk's above-paired

violin plots show the results of pairwise tests derived from LMMs comparing each acoustic parameter between deaf and control vocalizers, where ***$p < 0.001$, **$p < 0.01$, *$p < 0.05$ following Šidák correction. **c** Mean apparent vocal tract lengths (VTLs) with overlaid dot plots, derived from formant spacing, were shorter in deaf male vocalizers producing aggressive vocalizations than in controls (cont.), indicating a notable lack of vocal size exaggeration by deaf male vocalizers when expressing aggressive intent (LMMs **$p < 0.01$, †$p < 0.10$ Šidák corrected; $n = 140$ vocalizations; error bars represent standard errors of the mean, SEM). Below this, residual formant frequencies from a uniform relaxed vocal tract model closed at the glottis and open at the mouth, and an $F1$-$F2$ vowel space plot, further show that male vocalizers with typical hearing modulated individual formants (especially $F2$, but also $F3$ and $F4$) more than did deaf vocalizers, in each emotional context, indicating that deaf vocalizers produced comparatively unarticulated vocalizations.

including both vocalized and silent segments from the beginning to end of voicing) and the duration of vocalized segments relative to silent segments (dur Vocal). Unvoiced (e.g., breathy) vocal sounds were included as vocalized segments.

We manually measured nonlinear acoustic phenomena which are produced by aperiodic oscillations of the vocal folds and contribute to the rough and harsh acoustic quality of vocal signals[25,68]. Nonlinearities were identified and annotated from spectrograms (0–5 kHz; window length 0.05, see e.g., Fig. 1a) and from amplitude waveforms of voice recordings in *Praat* following standardized procedures for human vocalizations[19,20,69] and by two independent raters. For each vocalization, we computed the total durations of vocalized segments containing sidebands (durSide), subharmonics (durSub) or deterministic chaos (durChaos), from which we calculated the percentage of vocalized segments containing one or more nonlinear phenomena (%NLP). Twenty-eight of 360 vocal stimuli were marked by extreme deterministic chaos or were entirely unvoiced (e.g., gasps), and thus did not contain measurable $f_o$ or a harmonic structure, in which case $f_o$ and frequency modulation parameters were not computed.

Finally, we manually measured the first four formant frequencies ($F1$–$F4$), resonances of the vocal tract, in a subset of 140 vocal segments in *soundgen*[65] using the formant_app function[70]. Because formants require a relatively dense harmonic structure to be measured reliably, we selected vocalizations with a mean $f_o < 400$ Hz. Moreover, because too few female vocalizers produced such relatively low-pitched vocalizations, our formant analyses focused on male vocalizers. Formant measures were taken from

each independent call within a vocal recording. Calls are defined as independent voiced segments separated by at least 150 ms of aspiration or silence, sometimes accompanied by a change in vowel quality. We excluded fully closed-mouth vocalizations due to nasal formants. In total, we measured formants from 43 aggressive calls (42% by deaf male vocalizers), 51 pain calls (51% by deaf vocalizers), and 46 fear calls (37% by deaf vocalizers). Spectrograms were visually inspected to verify the fit of formant tracks to spectral peaks, followed by manual adjustment of LPC spectral smoothing and visual inspection of vowel quality, formant spacing ($\Delta F$), and apparent vocal tract length (VTL) to ensure a robust fit (see ref. 70). The measures $\Delta F$ and VTL were computed for each vocalizer based on $F1$–$F4$ following the validated regression method[70–72].

For use in perception experiments, each vocal stimulus was bound by 500 ms of silence in *Praat*[64]. Stimuli did not need to be amplitude normalized for playback as microphone distance was standardized at the time of recording (150 cm), and thus, variation in amplitude remained potentially informative.

## Listeners

A total of 444 participants representing a broad age range (16–60 years old) and a roughly even sex ratio (59% self-reporting as female) took part in four independent psychoacoustic perception experiments (listener sample descriptives are given below and in Supplementary Table 3). Sample sizes were pre-determined prior to experimentation: for forced-choice perception experiments (experiments 1,3,4), we created a stopping rule for data

collection at approximately 30 raters (15 male and 15 female raters) per voice stimulus, based on evidence that 15 raters per sex produces Cronbach's alphas exceeding 0.8 indicating a high degree of inter-rater agreement and consistency in ratings[73,74]. Due to random sampling for playback, some voice stimuli reached (and thus exceeded) 30 ratings earlier than did others; data were collected until minima were reached for all voice stimuli. For perception experiment 2, in which participants provided open-ended responses, the stopping rule was set to 50 listeners for a representative sample. Listeners were recruited from the general population and local community via online advertisements, posters and professional contacts.

### Perception experiments

Perception experiments were conducted in a quiet lab room in independent private sessions. Experimental sessions were launched by a research assistant using a custom computer interface in which instructions were presented on screen, and voice stimuli were presented through Sennheiser HD-280 Professional headphones. Before the experiment commenced, listener sex and age were recorded, and a comfortable listening level was determined via a short demo; this volume was then maintained at a constant level within listeners for the remainder of the study. For each participant, a computer algorithm randomly selected one of four perception experiments (see below) and then randomly selected voice stimuli from 20 deaf vocalizers and 20 typically hearing controls, maintaining a balanced sex ratio. All three vocalizations produced by each randomly drawn vocalizer (aggression, fear, pain) were then used as playback stimuli in that given experimental session. Hence, in each experiment, each listener judged a total of 120 vocalizations drawn from the full set of 360 vocal stimuli. Trials progressed automatically after each response. Voice stimuli were blocked by vocalizer sex; the order of blocks and the presentation of voice stimuli within each block were randomized such that vocalizations produced by deaf vocalizers and controls were intermixed within blocks.

Four perception tasks were conducted with four independent samples of listeners (Supplementary Table 3) who were instructed that they would hear a series of emotional vocalizations produced by male and female vocalizers in various contexts:

1. **Forced-choice emotion classification**. Listeners ($n = 139$, aged 16–55) were instructed to choose from three possible emotional contexts (aggression, fear, pain), indicating in which context a given vocalization was most likely to have been produced. The exact same scenarios given to the vocalizers (see Supplementary Methods) were provided here to listeners. Context order was randomized between participants.
2. **Open-ended emotion identification**. Listeners ($n = 51$, aged 18–52) were instructed to indicate, in an open-ended one-word response, the intended emotion of a given vocalization. No specific information was provided about possible emotional contexts (i.e., aggression, fear, or pain were not mentioned).
3. **Authenticity identification**. Listeners ($n = 117$, aged 18–59) were provided with the intended emotion (aggression, pain, or fear) of each vocalization they heard, and were instructed to indicate how authentically each vocalization expressed that emotion. Ratings were given on a scale from 1 (not at all authentic) to 7 (completely authentic). Emotion context order was randomized between participants.
4. **Deafness detection**. Listeners ($n = 137$, aged 16–60) were told that the vocalizations they would hear were produced by men and women, some of whom were typically hearing and others who were profoundly deaf. In a two-alternative forced-choice task, for each vocalization they heard, listeners were instructed to indicate whether that given vocalizer had normal hearing or was hearing-impaired.

### Statistics and reproducibility

Data analysis was performed using R (version 4.3.2) and SPSS (version 25). We ran a series of linear mixed models (LMMs) fit by restricted maximum-likelihood estimation to examine differences in the acoustic structures and perceptual qualities of vocalizations produced by deaf vocalizers versus typically hearing controls. Full parameters for omnibus and final models are

detailed in the footnotes of each respective output table in the Supplementary Tables. Significant effects in LMMs were further examined using pairwise tests with Šidák correction for multiple comparisons. All tests were two-tailed with an alpha of .05. To test for acoustic atypicality, we ran LMMs for each individual voice parameter split by sex of vocalizer and emotional context (see Supplementary Tables 1, 2 for descriptive statistics of voice parameters). The key variable of interest (deaf or control vocalizer) was entered in all models as a fixed variable, and the anonymous ID of vocalizers was included as a random variable with random intercepts.

We performed principal component analysis (PCA) with varimax rotation on all voice parameters to extract a reduced number of uncorrelated acoustic parameters[75]. The PCA classified the 15 voice parameters into 5 clear components, together explaining 83% of the variance across vocalizations (PCA Supplementary Table 4). The five principal components respectively represented: voice pitch mean and range ($f_o$ mean, min, max); amplitude (AMP mean, max, and IntCV); noise and perturbation parameters (HNR, jitter, shimmer, with some influence of nonlinear vocal phenomena, NLPs); duration (Dur, Dur Vocal); and frequency modulation (inflex2 and inflex25). Factor scores from all 5 principal components were then inputted into leave-one-out discriminant function analyses (DFA) with forced entry, as this method is less susceptible to collinearities, type I errors, and random effects[76]. The DFAs were used to test for acoustic distinctiveness in vocalizations across emotion contexts (aggression, pain, fear), producing cross-validated correct classification percentages for each emotion and group of vocalizers (DFA Supplementary Table 5).

To analyse listeners' responses in perception experiments, ratings were coded numerically in experiments 1, 3, and 4. We additionally computed the proportion of correct responses in experiments 1 and 4. These coded responses were entered as the dependent variables in a series of LMMs. The key variable of interest (deaf or control vocalizer) was entered as a fixed variable, and the anonymous IDs of both vocalizers and listeners were included as random variables with random intercepts to control for individual differences and avoid pseudo-replication. The sex of both vocalizer and listener were also entered as fixed variables in omnibus models. Vocalizer sex consistently showed significant effects in omnibus models, therefore separate LMMs are typically reported for male and female vocalizers, except for models comparing listeners' judgments between the most and least severely deaf vocalizers due to reduced sample size. In contrast, omnibus models did not consistently show significant main or interaction effects of listener sex, therefore listener data were pooled for all analyses.

Data distributions in linear mixed models were assumed to be normal, but this was not formally tested. Thus, in addition to these LMMs, we ran analogous binomial generalized linear mixed models (GLMMs) with logistic link functions on the binary response data from experiments 1 and 4, wherein responses were now coded as correct or incorrect (0,1) rather than as proportions. These GLMMs produced the same results as LMMs, and both are reported in the Supplementary Tables. We additionally computed unbiased hit rates (Hu scores[77]) from emotion classification responses to control for apparent response biases. Hu scores were arcsine-transformed for use in paired sample Wilcoxon signed-rank tests. Finally, data from the open-ended emotion identification experiment consisted of a total of 6120 single-word responses collected from 51 participants, each labeling 120 vocalizations. Two researchers independently and blindly coded the valence of each response as negative, neutral/ambiguous, or positive. Inter-rater agreement was high (Cohen's $\kappa = 0.82$) and any discrepancies were discussed, agreed upon and revised as necessary (see Supplementary Table 6).

### Reporting summary

Further information on research design is available in the Nature Portfolio Reporting Summary linked to this article.

## Results

### Deaf vocalizers produced acoustically atypical vocalizations

Our first aim was to test for differences in the underlying spectrotemporal acoustic structures of vocalizations produced by deaf adults compared to

typically hearing controls, with key results summarized in Fig. 1. We focused on 15 voice parameters known to be important in nonverbal vocal communication, including fundamental frequency ($f_o$, perceived as pitch) and its variability, vocal perturbation and noise, amplitude, temporal parameters, and the proportion of harsh nonlinear acoustic phenomena (see Supplementary Tables 1, 2 for acoustic measures). Discriminant function analyses of acoustic components (see Supplementary Table 5 for DFA), derived from a principal component analysis of all 15 voice parameters (see Supplementary Table 4 for PCA), showed that the vocalizations of aggression, fear, and pain produced by deaf vocalizers overlapped significantly in acoustic space (Supplementary Fig. 1), more so than did those of hearing controls. For example, the DFA correctly classified aggressive vocalizations in deaf adults with only 37% accuracy, compared to 58% accuracy in hearing controls. This shows that the vocalizations of deaf adults were relatively more homogeneous in their acoustic structures regardless of their intended emotion.

Linear mixed models (LMMs) further showed structural atypicality in each type of vocalization produced by deaf adults (Supplementary Tables 7–9 for full LMMs), especially aggressive vocalizations (Supplementary Table 7). As predicted, control vocalizers with typical hearing produced aggressive vocalizations with a relatively low pitch, high perturbation, and a high proportion of nonlinear phenomena. This acoustic structure is common in agonistic calls because their ostensible function is to threaten[2,5,26]. In contrast, deaf vocalizers produced aggressive vocalizations that were unusually tonal with significantly less jitter ($F_{1,58} = 7.8$, $p = 0.007$), higher harmonics-to-noise ratios (HNRs, $F_{1,58} = 24.3$, $p < 0.001$), and most notably, six times fewer nonlinear phenomena on average (NLP, $F_{1,58} = 38.6$, $p < 0.001$; $7.5 \pm 4.3\%$ male vocalizers, $9 \pm 4.7\%$ female vocalizers) than those of controls ($45.3 \pm 4.3\%$ male vocalizers, $40.5 \pm 4.7\%$ female vocalizers; Fig. 1b and Supplementary Table 7). The aggressive vocalizations of deaf vocalizers thus critically lacked a characteristic harshness and roughness. Deaf male vocalizers also produced aggressive vocalizations that were, on average, 100 Hz higher pitched (mean $f_o$ $451 \pm 30$ Hz) than were those of male controls ($349 \pm 33$ Hz; Supplementary Tables 1, 7 and Fig. 1b).

We found similar patterns of results for pain vocalizations. These contained a moderate proportion of harsh nonlinear phenomena among typically hearing male ($36 \pm 3.7\%$) and female vocalizers ($15 \pm 3.1\%$), yet almost no nonlinearities among deaf males ($4.9 \pm 3.7\%$) and deaf female vocalizers ($3.3\% \pm 3.1$, Supplementary Table 8 and Fig. 1b). Thus, like for aggression, the pain vocalizations of deaf adults had significantly fewer nonlinearities than did those of hearing adults (male vocalizers: $F_{1,58} = 36.4$, $p < 0.001$; female vocalizers: $F_{1,58} = 7.2$, $p = 0.01$) and were in fact almost entirely devoid of harshness. Deaf male vocalizers also produced relatively more tonal (higher HNR) pain vocalizations ($F_{1,58} = 5.8$, $p = 0.019$), and while differences in amplitude were marginally nonsignificant (mean AMP, $F_{1,58} = 3.8$, $p = 0.057$), the pain vocalizations of deaf male vocalizers were slightly quieter then were those of male controls (Supplementary Table 8). Aggressive and pain vocalizations were thus unusually tonal and high-pitched in deaf vocalizers compared to our sample of typically hearing humans (and other healthy human samples[20,26,29]), and more broadly, compared to other healthy mammals[25,68].

Unlike aggression and pain, we found that fear vocalizations were structurally similar between deaf and control vocalizers, especially among female vocalizers (Supplementary Table 9 and Fig. 1b). Both groups produced relatively tonal fear calls with a comparably low proportion of nonlinear phenomena (%NLP 4–11%) that did not differ significantly between deaf and hearing male ($F_{1,58} = 0.1$, $p = 0.799$) or female vocalizers ($F_{1,58} = 2.7$, $p = 0.08$). However, deaf male vocalizers produced fear vocalizations that were even more tonal and harmonic than were those of hearing male controls (lower jitter, $F_{1,58} = 4.7$, $p = 0.034$; higher HNR, $F_{1,58} = 8.9$, $p = 0.004$). Most notably, deaf male vocalizers produced relatively higher pitched fear vocalizations (mean $f_o$ $446 \pm 34$ Hz) than those of hearing male controls (mean $f_o$ $323 \pm 37$ Hz, Fig. 2b; $F_{1,58} = 6.0$, $p = 0.018$), thus producing 'hyper' versions of stereotypical distress calls.

While the durations of calls did not differ between vocalizer groups (dur, dur Vocal: Tables S1, S2 for acoustic parameters, LMMs Supplementary Tables 7–9), deaf vocalizers produced calls with more individual segments (mean unvoiced breaks 1.44, max 8) than did controls (mean 1.12, max 4) in all three emotional contexts and by both sexes. The vocalizations of deaf adults were thus generally less temporally fluid, containing more pauses and sudden breaks.

The production of distinctive call types like fear screams versus aggressive roars can also involve articulatory maneuvers and vocal tract configurations that influence formant frequencies, resonances of the vocal tract. Specifically, compared to fear, aggressive calls in humans[78] and other mammals[28] typically have denser formant spacing signaling a longer apparent vocal tract length that functions to maximize impressions of body size and formidability[78,79]. This can be achieved either by lowering the larynx in the vocal tract and reducing overall formant spacing, or by manipulating the articulators, such as by protruding the lips, thus changing the relative positions of the lower formants while also affecting vowel quality[78]. In a sub-sample of vocalizations ($n = 140$ calls where $f_o < 400$ Hz, see Methods), linear mixed models showed that deaf male vocalizers produced vocalizations with a wider formant spacing ($F_{1,35} = 12.1$, $p = 0.003$) and thus with relatively shorter vocal tract lengths ($F_{1,34} = 10.7$, $p = 0.003$) compared to controls in the aggressive context (see Supplementary Tables 10, 11 for full LMMs). Indeed, hearing controls extended their vocal tracts by nearly 2 cm more than did deaf vocalizers when trying to sound aggressive (Fig. 1c), whereas deaf vocalizers did not exaggerate their body size by lowering their formants (nor their voice pitch, as noted above). Such group differences in formant spacing and apparent vocal tract length were less evident in the pain context and were entirely absent in fear vocalizations, where selection pressure to sound large is arguably weaker (Supplementary Tables 10, 11 and Fig. 1c).

We found that male vocalizers with typical hearing achieved narrower formant spacing (and thus longer vocal tracts) by lowering their upper formants ($F3$ and $F4$), specifically in the context of aggression (Supplementary Table 12 for full LMM). Similar formant modulation has been observed in other healthy human samples[78–80] and other animals[4] to exaggerate body size and communicate threat. In contrast, we found that deaf male vocalizers *raised* their upper formants when instructed to sound aggressive ($F3$: $F_{1,35} = 8.7$, $p = 0.006$; $F4$: $F_{1,33} = 6.1$, $p = 0.019$), resulting in a shorter apparent vocal tract that is known to convey the impression of a relatively smaller body size (Fig. 1c). In addition, male vocalizers with typical hearing consistently produced vocalizations with a significantly lower second formant across all emotional contexts ($F2$: $F_{1,44} = 25$, $p < 0.001$), and thus a smaller gap between the first and second formants compared to deaf male vocalizers (Fig. 1c). Smaller $F1$-$F2$ spacing typically characterizes sounds produced with rounded lips such as "oo", whereas wider $F1$-$F2$ spacing can indicate spreading of the lips such as during the production of "ee", with the former more likely to signal threat and the latter more closely related to appeasement in animal communication. By plotting residual formant frequencies from those predicted by a uniform relaxed vocal tract[71], we show that the observed difference in $F2$ between deaf and hearing vocalizers is due to a general lack of modulation by deaf vocalizers. Indeed, the second and upper formants deviated less from their default positions in deaf vocalizers than in controls (see Fig. 1c), suggesting that deaf vocalizers produced relatively unarticulated vocalizations with minimal manipulation of the lips and tongue and with a relaxed vocal tract[81].

In summary, our acoustic analyses revealed structural anomalies in the emotional vocalizations of deaf adults, who produced unusually tonal, high-pitched, and unarticulated calls regardless of their intended emotion. This shows that some emotions and motivations, especially aggression, are not stereotypically encoded in the calls of people with deafness. So, can they be decoded? To answer this question, we next tested how these vocalizations are perceived by human listeners.

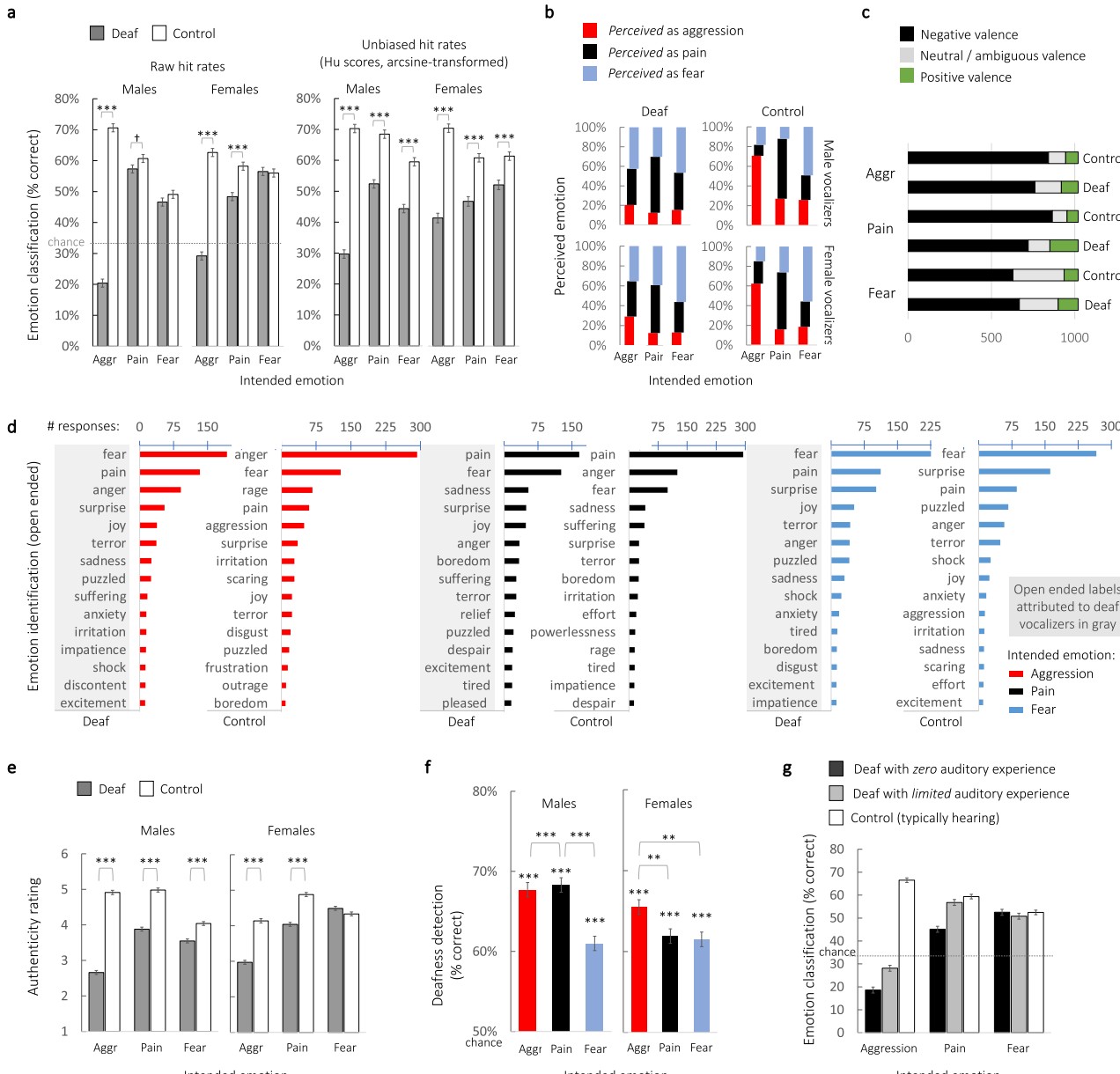

**Fig. 2 | Listeners struggled to identify the intended emotions and valence of vocalizations by deaf adults, perceived them as least authentic, and readily detected deafness. a** Accuracy in emotion classification in a forced-choice task (experiment 1, $n = 139$ listeners) showing that listeners were better at identifying aggression and pain from typically hearing control vocalizers (white bars, $n = 60$) than from deaf vocalizers (gray bars, $n = 60$) based on both raw hit rates and unbiased hit rates (Hu scores, arcsine-transformed). Unbiased hit rates and paired sample Wilcoxon signed-rank tests ($p < 0.001$) further showed that, when individual biases were controlled, listeners were relatively worse at judging all three emotions in deaf vocalizers than in controls. **b** Error distributions in perceived emotions (y-axis, see legend) plotted as a function of the vocalizer's intended emotion (x-axis), show that aggression was often confused as fear followed by pain in deaf but not control vocalizers (experiment 1, $n = 139$ listeners). **c** Valence of one-word responses used to describe vocalizations in the open-ended emotion identification task (experiment 2, 6120 one-word responses from $n = 51$ listeners), where black indicates negative valence, gray indicates neutral or ambiguous valence, and green indicates positive valence. Listeners erroneously attributed a higher proportion of positive or neutral valence to the deaf than control vocalizers for each intended emotion. **d** The top 15 emotion labels attributed to vocalizations based on their intended emotion, for deaf vocalizers (words in gray boxes) and control vocalizers, plotting the number of responses for each word (experiment 2, 6120 one-word responses from $n = 51$ listeners). Words attributed by

listeners to the vocalizations of typically hearing vocalizers were more congruent with the actual intended emotions than were those attributed to deaf vocalizers, particularly in the aggressive context, in which the labels "fear" and "'pain" were erroneously used most often to describe aggressive vocalizations produced by deaf vocalizers. See also Fig. S2 for word clouds. **e** Mean authenticity ratings (experiment 3, $n = 117$ listeners) were lower for vocalizations produced by deaf (dark bars) than control (white bars) vocalizers in all cases except female fear vocalizations, where 1 indicates "not at all authentic" and 7 indicates "completely authentic". **f** Mean percentages of correct deafness detection (experiment 4, $n = 137$ listeners) show that listeners could often identify deaf vocalizers from their nonverbal vocalizations. Comparisons against chance (50%) derive from one-sample $t$-tests, where $***p < 0.001$. **g** Mean percentages of correct classification of aggression and pain (experiment 1, $n = 139$ listeners) were lowest for vocalizations produced by congenitally deaf adults with zero auditory experience (no history of hearing aid or cochlear implant, $n = 20$ vocalizers, black bars) followed by deaf vocalizers with some limited prior acoustic experience ($n = 40$ vocalizers, gray bars) and finally, for typically hearing controls ($n = 60$ vocalizers, white bars), where the dotted line indicates chance (33%). In all panels, estimated marginal means and pairwise comparisons derive from LMMs unless stated otherwise, where $***p < 0.001$, $**p < 0.01$, $*p < 0.05$ following Šidák correction. Error bars represent standard errors of the mean, SEM. When colors are present, red always indicates aggression (aggr), black indicates pain, and blue indicates fear.

## Nonverbal vocalizations of deaf vocalizers are perceived atypically by listeners

Having shown that the vocalizations of deaf adults were acoustically atypical, next, we tested whether listeners perceived these vocalizations differently than those of typically hearing vocalizers. Our key results are summarized in Fig. 2. Four independent samples totaling 444 adult listeners who reported typical hearing (59% female, aged 16–60; Supplementary Table 3) judged subsets of the 360 vocalizations produced by deaf and control vocalizers. Listeners were randomly assigned to one of four lab experiments: (Exp. 1) Forced-choice emotion classification; (Exp. 2) Open-ended emotion identification; (Exp. 3) Authenticity identification; and (Exp. 4) Deafness detection (see Methods). Based on our acoustic analyses, we predicted that listeners would more often misidentify the intended emotions of deaf vocalizers compared to hearing controls, would judge their vocalizations as least authentic, and would be capable of detecting hearing impairment from nonverbal vocalizations alone, owing to the atypical acoustic structure of those produced by deaf vocalizers.

Mixed models based on raw hit rates confirmed that listeners ($n = 139$ exp. 1) correctly classified the vocalizations of control vocalizers by their intended emotion, far exceeding chance (33%) with an average accuracy of 60% ± 0.005 SEM (Fig. 2a). This was significantly higher than the emotion classification for deaf vocalizers (43% ± 0.005; Supplementary Table 13a, b for full models). This group difference was, however, largely driven by aggressive vocalizations, in both male ($F_{1,2796} = 944.9$, $p < 0.001$) and female vocalizers ($F_{1,2808} = 348.8$, $p < 0.001$). Indeed, while listeners correctly identified aggression in male controls (70.6 ± 1.2% correct) and female controls (62.4 ± 1.2%), they could not correctly identify aggression in deaf male (20.4 ± 1.2%) nor deaf female vocalizers (29.1 ± 1.3%), where accuracy fell well below chance (Fig. 2a). While pain was also more difficult for listeners to recognize in deaf vocalizers compared to controls, particularly in female vocalizers ($F_{1,2768} = 27.0$, $p < 0.001$), listeners correctly identified fear with the same veracity for all vocalizers regardless of whether they were deaf or hearing, when only raw hit rates were considered (Fig. 2a left panel; confirmed with TOST equivalence tests[82]: 90% CIs for Cohen's $d = -0.44$ to 0.42 in female vocalizers, and $-0.37$ to 0.48 in male vocalizers, fell within equivalence bounds of $d = -0.50$ to 0.50).

As the above comparisons were based on raw hit rates, the analyses did not take response biases into consideration. Such biases were evident from confusion matrices examining errors in listeners' judgments (Fig. 2b). These confusion matrices showed that the aggressive vocalizations of deaf adults were very frequently confused with fear (42/35% male/female vocalizers) followed by pain (37/11% male/female vocalizers). Listeners were thus biased toward perceiving vocalizations produced by deaf individuals as fearful, across all emotional contexts. We therefore computed unbiased hit rates for each participant (Hu scores[77], Supplementary Table 14 and Fig. 2a right panel). Unbiased hit rates control for individual predispositions in listeners to select a given emotional context (e.g., fear) more often than any other. Wilcoxon signed-rank tests on arcsine-transformed Hu scores showed that listeners were, in fact, significantly more accurate in classifying all emotions from the vocalizations of typically hearing adults than from those of deaf vocalizers when taking into account such biases (all $Z > 5.8$, all $p < 0.001$, Supplementary Table 15 and Fig. 2a right panel). This shows that by overestimating fear in deaf vocalizers, listeners were relatively more accurate when classifying fear, at the expense of correctly classifying aggression, but were still significantly worse at classifying fear and all other emotional contexts from the vocalizations of deaf adults.

Similar patterns of results were observed in the open-ended emotion identification task ($n = 51$ exp. 2). Listeners used hundreds of different words to describe vocalizations but were more likely to correctly attribute appropriate labels to the vocalizations of typically hearing controls than to those of deaf vocalizers, again, especially for aggression (Fig. 2c, d). Figure 2d shows the top 15 words that listeners attributed to vocalizations based on their intended emotion and the vocalizer group (see also Supplementary Fig. 2 for word clouds and Supplementary Table 6 for the full list of one-word responses). For typically hearing controls, we show that listeners

correctly attributed the labels 'anger', 'pain', and 'fear' to the appropriate vocalizations most of the time, accounting for nearly 300 responses in each emotional context. In contrast, for deaf vocalizers, listeners attributed far fewer appropriate labels to each intended emotion. In the case of aggressive vocalizations, they were much more likely to misattribute labels such as 'fear' (193 responses) or 'pain' (133 responses) rather than 'anger' (91 responses) or 'aggression' (7 responses) to deaf vocalizers (Fig. 2d and S2). Indeed, the word 'anger' was used three times more often to describe the aggressive vocalizations of controls (293 responses) than those of deaf vocalizers (91 responses).

The open-ended emotion labels attributed to deaf vocalizers were also more likely to be positively valenced such as 'joy', which was in the top five words attributed to deaf vocalizers across all three emotional contexts, or 'excitement', 'pleased' and 'relief' which were in the top 15 words (Fig. 2d). Because all three emotional contexts were negative, this showed that listeners made more mistakes not only in identifying the intended emotions of vocalizations produced by deaf adults, but even their negative valence. By additionally classifying words by their valence (Fig. 2c), we show quantitatively that listeners were more likely to incorrectly attribute positive valence (e.g., joy, excitement, relief) and neutral or ambiguous valence (e.g., puzzled, surprised, effort) to the vocalizations of deaf vocalizers than to those of controls (see Fig. 2c and Supplementary Table 6).

The vocalizations of deaf adults were also perceived as significantly less authentic than those of controls, as judged by another sample of listeners ($n = 117$ exp. 3; Supplementary Table 16 for full LMM). Linear mixed models confirmed that this was true for all three emotions and for both vocalizer sexes (all $F_{1,2338} > 36.5$, all $p < 0.001$), except female fear vocalizations where this group difference was marginally nonsignificant ($F_{1,2338} = 3.8$, $p = 0.052$, Fig. 2e) as confirmed with TOST equivalence tests[82] (90% CIs for Cohen's $d = -0.50$ to 0.35 fell just within equivalence bounds of $d = -0.50$ to 0.50). Aggressive calls produced by deaf adults were judged as least authentic of all, especially those of deaf male vocalizers which were judged as nearly half as authentic ($M = 2.7 ± 0.06$ SEM on a scale of 1 to 7) as were those of male controls ($M = 5.0 ± 0.06$; Supplementary Table 16 and Fig. 2e). This corroborates the results of the above emotion identification experiments, together showing that listeners cannot identify aggression in deaf vocalizers, and do not perceive their vocalizations as realistically expressing aggression.

A fourth and independent group of listeners ($n = 137$ exp. 4) correctly identified hearing impairment in deaf vocalizers on average 64% ± 0.004 of the time (Fig. 2e), exceeding chance (50%) across emotional contexts and vocalizer sexes ($t$-tests, all $t > 11.7$, all $p < 0.001$). However, mixed models showed a significant effect of emotional context in deafness detection for both male vocalizers ($F_{2,5478} = 24.5$, $p < 0.001$) and female vocalizers ($F_{2,5478} = 6.7$, $p = 0.001$; Supplementary Tables 17a, b for full models). Pairwise comparisons following Šidák correction confirmed that listeners were more successful in detecting deafness from aggressive vocalizations in both male (67.7 ± 0.01%) and female vocalizers (65.5 ± 0.01%), than from fear vocalizations in both male (61 ± 0.01%) and female vocalizers (61.5 ± 0.01%; all $p < 0.01$, Fig. 2f). This is likely because fear vocalizations sounded relatively typical in deaf vocalizers, whereas aggressive vocalizations sounded unconventional, as evidenced by our acoustic analyses and the other perception experiments.

Taken together, the results of these four perception experiments show that the vocalizations of adults with profound hearing loss are (i) often confused by listeners as expressing the wrong emotion and even the wrong valence, (ii) sound relatively inauthentic, and (iii) can often be identified as produced by a deaf person, even in the absence of linguistic content. All these effects were especially pronounced for aggressive calls, followed by pain, and were least evident for fear—matching what we found in our acoustic analyses.

If auditory input through the ears is required for aggressive and pain vocalizations to mature typically, as our results suggest, then we can expect that more sensory deprivation may lead to more atypicality in these call types. If so, hearing-impaired vocalizers with the least amount of auditory experience in their lifetimes are expected to produce vocalizations that are

the least stereotypical, and thus, would most often be misclassified as expressing the wrong emotion, would be perceived as least authentic, and would most often be detected as produced by a deaf vocalizer. To test this, we categorized deaf vocalizers into two groups: congenitally deaf adults who had never had a hearing aid or cochlear implant and thus had never heard a sound from the outside world in their lifetimes (i.e., those with zero hearing experience, $n = 20$), and deaf adults with some limited auditory experience, namely those who lost their hearing shortly after birth and/or had used a hearing aid or implant ($n = 40$, see Table 1). We then compared listeners' ratings of vocalizations produced by these two groups of deaf vocalizers versus hearing controls.

While listeners correctly identified aggression an average 67% of the time in typically hearing vocalizers based on raw hit rates, accuracy fell to 28% ($\pm 0.010$) for deaf vocalizers who had some prior auditory experience and fell further to 18% ($\pm 0.014$) for deaf vocalizers with zero auditory experience (Fig. 2g). Thus, the more severe the hearing impairment of the vocalizer, the less effectively they communicated aggressive intent ($F_{1,2795} = 30.8, p < 0.001$, Supplementary Table 18 for full models). Listeners were also the worst at identifying pain from deaf vocalizers with no auditory experience (45% $\pm 0.016$) relative to those with some (57% $\pm 0.012$; $F_{1,2780} = 35.4, p < 0.001$). However, corroborating our acoustic analyses, fear identification was similar for all vocalizers regardless of whether they had typical hearing (52.5% $\pm 0.016$), limited hearing experience (50.8% $\pm 0.012$), or zero hearing experience (52.5% $\pm 0.016$; Fig. 2g and Supplementary Table 18 for full models; confirmed with TOST equivalence tests comparing the two deaf groups[82]: 90% CIs for Cohen's $d = -0.48$ to 0.42 fell within equivalence bounds of $d = -0.50$ to 0.50). We further found that listeners judged aggressive vocalizations ($F_{1,2338} = 4.5, p = 0.034$) as least authentic when produced by deaf vocalizers with zero auditory experience (exp. 3 data, Supplementary Table 19), and most often correctly detected hearing impairment from their aggressive ($F_{1,2738} = 8.3, p = 0.004$) and fear ($F_{1,2738} = 20.3, p < 0.001$) vocalizations (exp. 4 data, Supplementary Table 20 for full models).

These results confirm that the more severe a vocalizer's auditory deficit, the less typical their vocalizations sound. This strongly suggests that acoustic experience from the outside world plays a role in the formation of volitional nonverbal vocalizations by facilitating vocal learning and/or vocal motor control, particularly for vocalizations communicating aggressive intent.

## Discussion
Our results indicate that the rare human capacity to voluntarily express emotions using non-linguistic vocal signals like cries and roars may require auditory experience to develop typically. Indeed, many of the emotional vocalizations that we humans produce throughout our lifetimes are under our voluntary control, and this suggests that, like speech, these volitional vocalizations may need to be learned before they can be produced in a conventional, stereotypical manner. Here we combined acoustic analysis of nonverbal vocalizations produced by deaf and typically hearing adults with data from four perception experiments to provide converging support for this hypothesis. We show that deaf adults produce acoustically atypical vocalizations of aggression and pain. Moreover, listeners struggle to gauge the intended emotions of their vocalizations, perceive them as relatively inauthentic, and can detect deafness from their vocalizations alone. These results suggest that auditory input throughout the lifespan shapes the acoustic forms of some nonverbal vocalizations, namely aggression and pain, in turn allowing for the voluntary expression of these emotions and motivations. In contrast, a lack of auditory input and experience impedes this capacity. Sensory input throughout the lifespan from other modalities, such as vision, is thus clearly not sufficient for humans to develop the full repertoire of conventional volitional vocalizations whose acoustic forms are predictable and with which emotional intent can be effectively communicated to others. These results implicate vocal learning as a potential key player not only in the acquisition of speech, but also in the acquisition of non-linguistic human vocalizations, however a potential role of vocal motor experience cannot be excluded.

Our control group of healthy adults with typical hearing produced aggressive, pain, and fear vocalizations that differed acoustically from one call type to another in stereotyped and predictable ways, mapping onto the evolved communicative functions of each call type, not unlike the affective calls of other mammals[2]. For instance, in these control vocalizers, fear screams were predictably high-pitched and more tonal than were aggressive roars, which were low-pitched, harsh, and produced with an extended vocal tract. In contrast, we found that deaf adults produced vocalizations that were highly similar in their acoustic structures regardless of the emotion they were intended to convey, and as such, listeners could not easily tell them apart. Most notably, deaf adults produced aggressive and pain calls that were unusually high-pitched and tonal, with wide formant spacing and a lack of articulation. In turn, listeners often misidentified the intended emotion or valence of these vocalizations produced by deaf adults, judging a disproportionate portion of these calls as fearful. This corroborates predictions arising from conventional form-function mappings wherein high-pitched vocalizations are typically perceived as communicating fear or distress across animal species[5,83], with pitch also explaining the majority of the variance in fear perception from human screams[24].

Listeners also judged the vocalizations of deaf adults as less authentic compared to those of controls, except for the fearful calls of female vocalizers. Authenticity ratings can act as a proxy of how genuine or convincing a vocalization sounds to listeners in terms of conveying its intended emotion. Given atypicality in the acoustic forms of aggressive and pain vocalizations produced by deaf vocalizers, it is not surprising that they also sounded less authentic than did those of hearing adults. Authenticity ratings have been shown to positively predict perceptions of affective arousal and personal traits such as trustworthiness[58], broadening the social implications of such attributions. Listeners in our experiments also detected, well above chance, when vocalizations were produced by a person with hearing loss. This provides further converging evidence that listeners are sensitive to acoustic atypicality in the vocal signals of deaf adults, even when those signals are entirely non-linguistic.

The acoustic forms of vocalizations produced by hearing-impaired adults differed in many ways from those of hearing controls. Most notably, they lacked a "low and harsh" acoustic profile. Deaf vocalizers produced aggressive and pain vocalizations that contained several times fewer nonlinear phenomena than did those of controls, resulting in tonal instead of harsh calls. The aggressive calls of deaf male vocalizers were also unusually high-pitched, corroborating findings on speech prosody in deaf persons[84]. Moreover, unlike typically hearing controls, deaf vocalizers did not extend their apparent vocal tract lengths when conveying aggression, a vocal maneuver that exaggerates apparent body size and signals threat[78,79]. The fact that aggressive vocalizations produced by deaf persons did not sound like "typical" aggressive vocalizations can, in turn, explain why they were often confused with fear or pain by listeners. Human listeners are known to associate low voice pitch, narrow formant spacing, and the presence of harsh nonlinear phenomena with aggression, formidability, and large body size (refs. 2,56 for reviews). We have respectively coined these the 'low is large'[85] and 'harsh is large'[26] sensory biases. Tonal and high-pitched sounds are conversely associated with submission and distress[8,23,24]. Human infants as young as four months of age[86] and blind adult listeners[87,88], despite having limited to no visual experience, also show analogous perceptual correspondences suggesting that these sound-symbolic associations may not require a great deal of sensory experience to emerge in listeners and may even be at least partly innate. While Sauter and colleagues[55] did not measure nonlinear phenomena or acoustic harshness in the vocalizations of their eight hearing-impaired vocalizers, they found that anger vocalizations were significantly higher pitched in hearing-impaired than control vocalizers and likewise could not be correctly identified by listeners, consistent with our results.

While we found that deaf vocalizers did not produce vocalizations that were much quieter or louder, nor longer or shorter, than were those of typically hearing adults, their vocalizations had more unvoiced breaks and segments across all emotional contexts. Deaf individuals have been shown to produce more pauses in speech as well, including grammatically

inappropriate pauses, likely due to impoverished control over breathing or chest and abdominal movements during speech production[84]. Deaf individuals can also struggle to articulate specific vowel sounds in speech[84,89], likely due to relatively poor control over the articulators (lips, mouth, jaw, tongue) when speaking, wherein such control is necessary to achieve specific formant positions. Although shifting all formants down or lowering individual upper formants can communicate aggressive intent and exaggerate body size in humans[78,79] and other mammals[4,28], at the same time, changes in the lower formants and specifically $F2$ relative to $F1$ are closely linked to human speech sounds and depend largely on movements of the articulators that determine the perceived vowel quality of vocal output[81]. Our formant analyses show that deaf vocalizers did not lower their upper formants in contexts of aggression or pain and did not substantially modulate their second formant from its resting position, for any intended emotion. Modulation of $F2$ is typically achieved by rounding the lips or raising the back of the tongue to constrict the posterior oral cavity, as in the production of the /u/ ("oo") vowel sound. In contrast, raising $F2$ is achieved by constricting the space behind the lips, as with a smile, and in the production of /i/ ("ee")[81]. A relatively unmodulated $F2$, as observed in our deaf vocalizers, indicates a lack of articulation, as observed in the production of the neutral *schwa* vowel (ə). Our results thus suggest that the unusual vowel structures observed in the speech of deaf adults[45,46] can generalize to their non-linguistic vocalizations.

Unlike pain and especially unlike aggressive vocalizations, which we show are highly acoustically atypical in deaf vocalizers, fear vocalizations largely (but not wholly) appear to retain some of their normal acoustic structure and perceptual properties even in the absence of acoustic input from the external world. There are several plausible and non-mutually exclusive explanations for this. The first potential explanation is that fear vocalizations might be more 'hard wired' than those expressing aggression or pain. Although the development of specific call types in humans is poorly studied, with most research focusing on the ontogeny of cries and laughter[2], we know that human babies are not born 'roaring'. Because aggressive roars emerge later in human ontogeny than do distress vocalizations or cries, they may require more auditory and/or vocal motor experience to develop typically into adulthood. Indeed, the pain cries and fear screams produced by the deaf adults in our study may most closely resemble the innate distress vocalizations that infants, even congenitally deaf infants, produce immediately at birth. The caveat is that even late-emerging calls may have an innate pre-programmed structure[21].

A second perhaps more parsimonious explanation is that deaf vocalizers produced calls that just happened to sound like a typical fear call. Fear vocalizations are usually high pitched with more tonality and sparser formant spacing compared to aggressive vocalizations[22,24] as also supported by our data from typically hearing vocalizers. We show that in deaf vocalizers, *most* of their vocalizations have exactly this type of acoustic profile, regardless of the emotion they intend to communicate. Even aggressive vocalizations were relatively high-pitched and tonal in our sample of deaf adults. In other words, the 'default' vocalization type produced by deaf adults most closely resembles that of a fear or distress call. Perhaps, in the absence of auditory-motor experience, humans come to produce a high-pitched tonal and unarticulated vocalization type. This may explain why listeners perceived most vocalizations produced by deaf adults as fearful and could not reliably recognize their aggressive intent. This perceptual bias was apparent both in confusion matrices and in unbiased hit rates which showed that when response biases were taken into consideration and controlled, the intended emotions of vocalizations produced by deaf vocalizers were recognized significantly less accurately than were those of controls for every single emotion, from aggression to pain and fear.

A third possibility is that in terms of vocal production mechanisms, vocal maneuvers required to voluntarily recreate conventional aggressive vocalizations may demand more vocal learning and/or vocal production experience than do those characterizing fear vocalizations, if, for example, they are mechanistically more difficult to reproduce. For instance, producing an authentic-sounding aggressive vocalization "on demand" requires substantially lowering voice pitch, extending the vocal tract by lowering the larynx or protruding the lips, and setting the vocal folds into an aperiodic vibratory regime to elicit harsh nonlinear phenomena. In contrast, the high pitch, high amplitude, and tonality of fear screams can be achieved more easily, for instance, by just pushing excess air out of the lungs while keeping the vocal tract relaxed, which simultaneously increases amplitude and pitch[90]. The extent to which different call types can be more or less easily imitated and the role of vocal learning in this capacity has not been extensively studied and warrants further investigation.

By comparing congenitally deaf adults who had absolutely no experience with sound to deaf adults who experienced some limited sounds in their early ontogeny or via a hearing aid, we show that the more severe a person's auditory deficit, the more atypical their volitional vocalizations sound. This strongly suggests that auditory deprivation has "additive" effects on vocal production. However, multiple direct and indirect mechanisms may be at play. First and foremost, we cannot conclude whether deficits in vocal production are due to a lack of external auditory input from others or a lack of internal auditory feedback from the ears (i.e., hearing one's own voice). In experiments on nonhuman animals, isolating individuals from conspecifics can indicate whether auditory experience with the species-typical vocal repertoire is required for normal vocal development, whereas deafening animals early in ontogeny additionally shows the effect of lost internal auditory feedback from self-emitted vocalizations on vocal production later in life[34]. In humans, both auditory input and auditory feedback mechanisms appear to independently contribute to normal speech development[91]. A lack of auditory feedback to monitor vocal production is thought to be an especially important contributor to the impaired vocal control abilities observed during speech production in deaf people, including impaired control over voice fundamental frequency and articulation (ref. 84 for review). Nevertheless, directly comparing the relative contributions of auditory input and feedback mechanisms, and their potential mediating effects on vocal control, is often not possible in studies of humans with natural causes of deafness, where all these mechanisms are simultaneously affected.

Several scholars have suggested that differences in the emotive expressions of sensory-impaired versus typically developing persons may also arise for reasons other than obstructed auditory input or vocal learning[52,92]. For instance, infrequent vocal production may affect vocal fold response properties and oral muscle control[52], including poor coordination of laryngeal muscles and abnormalities in their tension that can directly affect voice pitch and its stability[89]. Infrequent use of the tongue or coordination of other articulators (such as the jaw and lips) can lead to physical changes that may affect articulation, and even lung capacity can differ in deaf versus hearing individuals, affecting control of breathing and thus rate and fluency of vocal signals[89]. To tease apart the relative contributions of auditory experience versus vocal production experience in the development of nonverbal vocalizations, researchers will need to measure not only the severity of hearing loss in deaf volunteers, but also their overall experience with vocal production, alongside speech articulation and vocal control tests. Finally, perceived social stigmas surrounding the production of audible sounds by deaf persons may also lead to behavioral inhibitions in everyday life that may affect vocal behavior[93].

The relative importance of unimodal versus multi-modal (e.g., audio-visual) encoding of sensory information in the production and perception of non-linguistic vocal signals also remains to be directly tested. Our results, based on deaf but sighted adults, suggest that visual cues alone (e.g., observing people's mouth movements and mouth shapes when expressing aggression or pain) do not suffice to fully replicate their vocal counterparts in the absence of auditory cues. Experiments that involve deaf-blind volunteers could help to confirm this. We also cannot exclude the possibility that deaf vocalizers encode emotion in their vocalizations in a systematic but different manner than do persons with typical hearing. For instance, rather than encoding aggression via harshness and lowered pitch, deaf vocalizers may encode aggression using an array of different acoustic patterns. However, our comprehensive acoustic analyses revealed few differences between call types expressing different emotions in our deaf sample, as most vocalizations shared a similar ('fear-like') acoustic profile, suggesting that this is not the

case. Such a unique encoding system would also not be particularly functional, given that listeners in our study were often not able to decipher the intended emotions of vocalizations produced by hearing-impaired vocalizers.

A final outstanding question is whether auditory-motor experience is needed to produce stereotypical *spontaneous* vocalizations, such as reflexive calls produced in real-life contexts. While our results show that intentional or posed vocalizations require auditory input and possibly also vocal learning to develop typically, as does speech which is likewise volitional, reflexive vocalizations triggered by real emotional experiences may not require prior learning. In other words, spontaneous emotional signals may develop reliably in humans without sensory input. If this were true, it would lend support to the hypothesis that a lack of opportunity for vocal learning is driving atypicality in the volitional vocalizations of deaf adults, as opposed to a lack of vocal production experience or physical anomalies in the vocal anatomy. Existing research on human facial expressions in blind adults supports this possibility[54], as does at least one study on spontaneous laughter[52]. Despite some degree of acoustic atypicality in the laughter of deaf adults when watching comedy films, these relatively reflexive laughs appear broadly species-typical in form and qualitatively similar to those of people without hearing impairments[52]. However, evidence is needed from a broader range of affective states, including spontaneous vocal bursts of aggression and pain. In the present research, we achieved a high level of internal validity by studying the production of discrete emotional call types in volitional contexts. Studying the vocal behavior of deaf adults in real-life situations such as competitive sports, childbirth, and sexual pleasure would have introduced a level of complexity in terms of interpreting the vocalizers' intended emotions and motivations. Nevertheless, such research will surely help us to know whether auditory-motor experience, which we show shapes voluntary human vocalizations, is also needed for their involuntary production.

## Limitations
In this study, we examined only negatively valenced affective states. With the exception of laughter, researchers have indeed largely focused on negative call types in humans such as screams of fear, roars of aggression, and cries of pain, with far fewer studies on the communication of positive affective states[2] (but see ref. 94 for review on positive vocalizations). In the comparative voice sciences, this negative bias likely originates from research on nonhuman animals. Vocal behavior in agonistic and distress contexts is most intensively studied due to its obvious evolutionary relevance, with strong selection pressures leading to salient form-function mappings in these call types (refs. 2,4,28,83 for reviews). More research is thus needed to test for form-function mappings in positively valenced call types in both typically hearing and hearing-impaired communities. It should be noted that producing emotional vocalizations on demand can feel embarrassing for some people, whether or not they are hearing-impaired, and this can potentially affect the degree to which posed vocal displays reflect their spontaneous counterparts. To alleviate this, all vocalizers in our study were left alone during the voice recording task, as privacy is known to reduce such inhibitions.

## Conclusions
Although human nonverbal vocalizations remain grossly understudied compared to speech, emerging research suggests that they are universal and share key similarities across human cultures, as they do across mammalian species[2]. Critically, our special ability as humans to voluntarily control their production can offer a rare insight into the evolution of speech, wherein volitional nonverbal vocalizations might represent the missing link between animal calls and human speech[3]. The advanced capacity in humans to control our voices raises the possibility that nonverbal vocalizations, which emerged before speech in our ancestral past, may be acquired through vocal learning as the results of this research indeed suggest. We show that in the absence of a lifetime of auditory-motor experience, deaf humans develop highly homogeneous non-linguistic vocalizations that essentially sound the same whether they are intended to communicate aggression, pain, or fear, and as such, listeners often misjudge their intended emotions. Our acoustic and perceptual data thus offer converging evidence that the voluntary production of

emotional vocalizations may require vocal learning. This finding supports our hypothesis that the capacity for vocal control, a key precursor of speech, may have emerged in our ancestral past before language during the production of affective non-linguistic vocalizations (see ref. 3 for further discussion). Given the importance of emotional expression in our everyday social lives, these results also have clinical and pedagogical relevance for developing tools that can aid communication in people with sensory deficits, ultimately improving interpersonal relations between the hearing and deaf communities.

## Data availability
All anonymized data were included as supplementary materials (Supplementary Data 1 and Supplementary Data 2) and on the Open Science Framework (https://doi.org/10.17605/OSF.IO/CJNME). Supplemental materials include audio voice recordings (for research purposes), quantitative measures pertaining to acoustic analyses of vocalizations, and coded listener responses from perception experiments.

## Code availability
Custom codes used for acoustic and statistical analyses, where applicable, are available on the Open Science Framework (https://doi.org/10.17605/OSF.IO/CJNME).

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

## Acknowledgements

This work was funded by a National Science Center grant (OPUS scheme # 2017/25/B/HS6/00561) to A.O., and by a French National Research Agency grant ("SCREAM" ANR-21-CE28000701) to D.R. and K.P. The funders had no role in study design, data collection and analysis, decision to publish, or preparation of the manuscript. We thank Piotr Kupczyk for programming the perception experiments and assisting with data collection and collation. We thank Sandra Chudak, Sara Dusza, Agata Głuszak, Paulina Idziak, Anna Janczak, Joana Malinowska, Justyna Płachetka, Michał Pieniak, Marta Rokosz, Maja Rybka, Michał Stefańczyk, Magda Susarska, and Lidia Wojtycka for assisting with data collection. We thank Marcin Masalski for sharing the hearing screening software and Andrey Anikin for developing *soundgen* and offering advice on measuring acoustic nonlinearities and formants. We thank Roza Kamiloğlu and Virgile Daunay for their assistance in computing Hu scores. Finally, we thank Gregory Bryant, Julia Simner, Andrey Anikin, and Nicolas Mathevon for offering helpful feedback on earlier versions of the manuscript.

## Author contributions

Katarzyna Pisanski co-designed the experiments and collected data, contributed resources and tools, analyzed the data, performed statistical modeling, created figures and plots, and wrote the paper. David Reby conceived the research question, co-designed the experiments, and provided critical input during writing. Anna Oleszkiewicz co-designed the experiments and collected data, contributed resources and tools, analyzed the data, and provided critical input during writing.

## Funding

## Competing interests

The authors declare no competing interests.
