## [Peer Review File · Communications Psychology]

17th Nov 23

Dear Dr Pisanski,

Thank you for your patience during the peer-review process. Your manuscript titled "Humans need auditory feedback to produce typical volitional nonverbal vocalizations" has now been seen by 3 reviewers, and I include their comments at the end of this message. They find your work of interest, but raised some important points. We are interested in the possibility of publishing your study in Communications Psychology, but would like to consider your responses to these concerns and assess a revised manuscript before we make a final decision on publication.

We therefore invite you to revise and resubmit your manuscript, along with a point-by-point response to the reviewers. Please highlight all changes in the manuscript text file.

Although you are expected to address all concerns raised by reviewers comprehensively, we ask you to pay particular attention to the following points:

Please provide justification for focusing on vocalisations with a negative valence (see Reviewer #1 & 2) and acknowledge the lack of auditory-motor experience as a potential contributing factor (see Reviewer #2).

Further, please respond to Reviewer #3's concerns about the appropriateness of the statistical approach, which may require adding an additional analysis.

Please ensure you follow our statistical guidelines when reporting statistics (<https://www.nature.com/commspsychol/submit/submission-guidelines#statistical-guidelines>). Please note in particular our requirements for the reporting and interpretation of null-results. Non-significant findings derived from null-hypotheses significance tests should be reported in full, but may not be interpreted. Where you interpret null results, this interpretation must be based on Bayes Factors or equivalence tests.

Please provide information on informed consent and IRB approval in the main manuscript.

Lastly, please ensure the order of your section is as follows: Introduction – Methods -Results – Discussion.

Please note that your revised manuscript must comply with our formatting and reporting requirements, which are summarized on the following checklist: Communications Psychology formatting checklist and also in our style and formatting guide Communications Psychology formatting guide .

Please use the following link to submit your revised manuscript, point-by-point response to the referees' comments (which should be in a separate document to any cover letter) and the completed checklist:

[link redacted]

Please do not hesitate to contact me if you have any questions or would like to discuss these revisions further. We look forward to seeing the revised manuscript and thank you for the opportunity to review your work.

Best regards,

Saloni Krishnan

Saloni Krishnan, PhD
Editorial Board Member
Communications Psychology
orcid.org/0000-0002-6466-141X

EDITORIAL POLICIES AND FORMATTING

Editorial Policy: Policy requirements (Download the link to your computer as a PDF.)

* TRANSPARENT PEER REVIEW: Communications Psychology uses a transparent peer review system. This means that we publish the editorial decision letters including Reviewers' comments to the authors and the author rebuttal letters online as a supplementary peer review file. However, on author request, confidential information and data can be removed from the published reviewer reports and rebuttal letters prior to publication. If your manuscript has been previously reviewed at another journal, those Reviewers' comments would not form part of the published peer review file.

* **CODE AVAILABILITY:** All Communications Psychology manuscripts must include a section titled "Code Availability" at the end of the methods section. In the event of publication, we require that the custom analysis code supporting your conclusions is made available in a publicly accessible repository; at publication, we ask you to choose a repository that provides a DOI for the code; the link to the repository and the DOI will need to be included in the Code Availability statement. Publication as Supplementary Information will not suffice. We ask you to prepare code at this stage, to avoid delays later on in the process.

* **DATA AVAILABILITY:**

All Communications Psychology manuscripts must include a section titled "Data Availability" at the end of the Methods section or main text (if no Methods). More information on this policy, is available at <http://www.nature.com/authors/policies/data/data-availability-statements-data-citations.pdf>.

At a minimum the Data availability statement must explain how the data can be obtained and whether there are any restrictions on data sharing. Communications Psychology strongly endorses open sharing of data. If you do make your data openly available, please include in the statement:

We recommend submitting the data to discipline-specific, community-recognized repositories, where possible and a list of recommended repositories is provided at <http://www.nature.com/sdata/policies/repositories>.

If a community resource is unavailable, data can be submitted to generalist repositories such as figshare or Dryad Digital Repository. Please provide a unique identifier for the data (for example a DOI or a permanent URL) in the data availability statement, if possible. If the repository does not provide identifiers, we encourage authors to supply the search terms that will return the data. For data that have been obtained from publicly available sources, please provide a URL and the specific data product name in the data availability statement. Data with a DOI should be further cited in the methods reference section.

REVIEWERS' EXPERTISE:

Reviewer #1: Acoustic analyses of voices, cognitive mechanisms of voice production

Reviewer #2: Acoustic analyses of voices, cognitive mechanisms of voice production

Reviewer #3: vocal evolution

REVIEWERS' COMMENTS:

Reviewer #1 (Remarks to the Author):

The current manuscript reports on the findings of an acoustic analysis and four behavioural experiments examining the perception and production of volitional non-verbal fear, aggression and pain vocalisations by deaf individuals. The authors conclude that non-verbal vocalisations produced by deaf individuals differ in their acoustic properties from non-verbal vocalisations produced by hearing individuals. Specifically, deaf vocalisations tended to be more harmonic and included less evidence for vocal roughness. Across 4 perceptual experiments, the authors confirm that listeners cannot recognise deaf vocalisations as accurately as they can recognise vocalisations produced by hearing individuals. This effect is most pronounced for aggression vocalisations, and can still be observed for pain vocalisations but is from what I can see much diminished or even absent for fear vocalisations.

Overall the study poses an interesting and well-motivated research question. The study design is clearly explained, analyses appear appropriate and the paper is overall clearly written. However, the paper was written in a very condensed way, such that at least one (Exp2) of the listening experiments is barely analysed/discussed, another is not motivated in the text (Exp4), and almost all analytical details related to the complex acoustic analysis are reported in the supplementary materials. Further, although this cannot be readily changed now, I unfortunately think that the insights we're getting from this paper are somewhat incomplete and believe that including more vocalisations would have made this study much more impactful.

Below I list some additional concerns about the design, presentation, and interpretation of the study.

1. Why were only aggression, fear, and pain vocalisations used? There are many other types of non-verbal vocalisations the humans produce and I cannot see a satisfactory explanation for this choice of vocalisations in the manuscript. For example, wouldn't it have been useful to also include positively or at least not clearly negatively valenced vocalisations?

More importantly: Is the conclusion the paper draws from the data likely to generalise beyond these particular vocalisations and can we really glean a full-enough picture from the data at hand? Different vocalisations behave in different ways across the analyses presented in the paper - with fear vocalisations being both in terms of perception and production similar across deaf and hearing producers. Furthermore, the introduction points out that, for example, some studies show that (admittedly spontaneous) laughter vocalisations produced by deaf individuals can be recognised as such. Is it thus not likely that the claim the authors make about auditory feedback being necessary only holds for some vocalisations? I feel like the paper could make this point much clearer.

2. Exp2: The free text data from Exp2 are potentially quite rich and I feel like there are more impactful ways of describing or reporting these data than is done at the moment. For now only a couple of examples are given of the relative frequency with which certain emotion labels were used across vocalisations and groups, alongside a basic valence assessment of the words. Have the authors, for example, considered plotting the responses as a word cloud to be included as a subpanel in a figure in the main text? I'm not insisting on this being done but I feel like currently the data from this experiment are only presented and discussed very superficially and in a way that is quite removed from the data, such that it makes me wonder why the authors choose this particular task.

3. Exp4: I was interested in why the measure of authenticity was included. It is not mentioned in the

introduction, the findings are not discussed at length or tied back to another literature. I'm not sure if the authors would like to tie this scale better into the main narrative or whether they might perhaps prefer to move this scale to the supplementary materials. However, at the moment without further background and context, I did not find the reporting of this scale particularly satisfying as a reader.

Related to this, when reading the prediction for the perceptual experiments, I was unsure where the prediction "Based on our acoustic analyses, we predicted that listeners would judge the vocalizations of deaf men and women as least authentic" came from. Perhaps a couple of lines of explanation behind the rationale of this prediction would be useful.

4. Very minor point for Figure 1B: I don't know if this is intentional or not but the individual data points (i.e. black dots) are not centered on the violin plots. If not intentional, the authors might want to align the dots with the violins.

Reviewer #2 (Remarks to the Author):

Based on production (n = 120) and perceptual (n = 444) studies, Pisanski et al. make the case that volitional emotional vocalizations require learning to develop typically. The focus is on nonverbal vocalizations expressing aggression, pain and fear, and the main findings are that (1) deaf adults produce vocalizations that are acoustically atypical, and that such vocalizations (2) are less efficient at communicating the intended emotions, are perceived as less authentic, and are detected as being produced by deaf speakers.

This is a very well-written and well-powered study, the methods are sound, the research question is timely and well-motivated, and the level of detail and rigor of the analyses are impressive. This work also represents a novel and significant contribution to the literature, which is why I recommend publication, even though I have suggestions for improvement that could be addressed in a revision:

- In the Introduction (lines 91-94), I suggest that the authors avoid the outdated and oversimplified view that the limbic system corresponds to the 'emotional' brain. Other regions and networks support emotional processes, at subcortical and cortical levels (for more nuanced accounts, see e.g., Pessoa, 2023, *Neuroscience and Biobehavioral Reviews*; Lindquist et al., 2016, *Cerebral Cortex*).
- The authors' argument is based primarily on the role of auditory input, and much less emphasis is placed on how deaf individuals also differ in vocal control, experience, and expertise (and on how such differences potentially contribute to the observed effects). It is likely that both lack of auditory input and differences in vocal production skills play a role, and it would be helpful to emphasize the two factors in a more balanced way, or to justify the preferential focus on perceptual aspects. Perhaps it would make sense to speak more of auditory-motor experience throughout the paper, and less of auditory input alone.
- Related to the previous point, finding that the effects are larger for vocalisers with less auditory experience is not strong evidence for the role of auditory input, because these individuals are likely to also have less vocal production experience. In fact, I was wondering whether the authors have any measure of speech/voice production experience/proficiency, which they could correlate with their measures of interest. If not, this could be discussed a potential avenue for future work.
- Can the authors explain why they decided to focus on aggression, pain and fear vocalisations, specifically? I was surprised that none of the vocalizations was positive. Issues of generalisability

could be discussed in more detail.

- Using the word 'authenticity' when all vocalisations are non-authentic reads like a misnomer. Would expressions like prototypical, typical, or representative be more appropriate?

- Recognition accuracy: do the findings replicate when accuracy rates are corrected for possible response biases, using for example unbiased hit rates, Hu scores (Wagner, 1993, Journal of Nonverbal Behavior)?

- I would expect recognition accuracy to be higher than 60% for typical vocalisations, considering that accuracy rates are usually high for nonverbal vocalisations (e.g., Laukka & Elflein, 2021, Emotion Review; Sauter et al., 2010, Quarterly Journal Experimental Psychology; Belin et al., 2008, Behavior Research Methods; Lima et al., 2013, Behavior Research Methods) and that there were only three options. Can the authors comment on that?

- Recordings: If I understood correctly, each vocaliser recorded only one vocalisation per emotion, without time for familiarisation with the task and rehearsal. In my experience, the production of vocalizations on command is perceived as an unusual task, often difficult, and for many speakers it takes time, both for them to understand what is required and to feel comfortable producing the sounds. The potential role of these factors might be even more important for deaf vocalizers – for them the task is likely to be more difficult. Can these confounds explain some of the observed effects?

Reviewer #3 (Remarks to the Author):

In their submitted manuscript, Pisanski, Reby, and Oleszkiewicz report the results of an extensive study of volitional nonverbal vocalizations of aggression, pain, and fear, focusing on differences in the production and perception of these vocalizations as produced by deaf vs. hearing vocalizers. Acoustic analyses clearly showed average differences in the qualities of the vocalizations. In four perceptual experiments, listeners judged emotion from the deaf-produced vocalizations significantly less accurately, and perceived those vocalizations as significantly less authentic, compared to hearing-produced vocalizations. Listeners were also significantly accurate at detecting whether a given vocalization was produced by a deaf vocalizer. The researchers report the effects of intended emotion and of vocalizer sex. The central conclusion, that voluntary production of typical emotional vocalizations requires vocal learning, is well supported by the data.

This study represents a substantial effort and carries the potential to significantly advance the field of nonverbal vocal communication. I read the manuscript with great interest, and carefully reviewed the researchers' experimental, acoustic, and statistical methods, results, and interpretation. I was impressed with the comprehensiveness of the research, the novel study conceptualization, and the thorough and well-articulated discussion. I have recommendations regarding statistical analysis and some minor revisions to incorporate additional relevant literature, as well as a question about acoustic variables.

Statistical Analyses

--My understanding is that in Experiments 1 and 4, the authors first computed each listener's proportion of accurate responses for each emotion category and vocalizer sex, then used that as the dependent variable in subsequent analyses (LMM). Rather than aggregating the data in this way, I believe it would be more powerful to use a binomial GLMM with individual responses, each coded as accurate or inaccurate, as the dependent variable.

Minor Comments

--LI. 85-87 "While the forms and functions of many human vocalizations appear homologous to those of other animals, there is one critical exception: humans can effortlessly, voluntarily produce vocal sounds unlike any other terrestrial mammal, including other primates (3,29)." This seems to imply that other species are incapable of volitional production of vocalizations. Such a view is represented in the literature, but does not reflect the consensus. It is clear that the human ability to modulate the voice far exceeds that of all other terrestrial mammals, but there is some evidence that other species can volitionally use vocalizations. See, e.g.:

Schel, A. M., Townsend, S. W., Machanda, Z., Zuberbühler, K., & Slocombe, K. E. (2013). Chimpanzee alarm call production meets key criteria for intentionality. *PLoS One*, 8(10), e76674.

Ghazanfar, A. A., Liao, D. A., & Takahashi, D. Y. (2019). Volition and learning in primate vocal behaviour. *Animal Behaviour*, 151, 239-247.

For a review, see: Schwartz, J. W., Engelberg, J. W., & Gouzoules, H. (2020). Evolving views on cognition in animal vocal communication: contributions from scream research. *Animal Behavior and Cognition*, 7(2), 192-213.

--L. 120: The most recent publication on vocal learning in nonhuman primates cited here is from 1997. I recommend incorporating more recent work on this subject, e.g.:

Egnor, S. R., & Hauser, M. D. (2004). A paradox in the evolution of primate vocal learning. *Trends in Neurosciences*, 27(11), 649-654.

Seyfarth, R. M., & Cheney, D. L. (2010). Production, usage, and comprehension in animal vocalizations. *Brain and Language*, 115(1), 92-100.

--Discussion: Regarding the finding that listeners perceived higher-pitched and tonal vocalizations as more fearful (including aggressive vocalizations produced by deaf individuals), it might be relevant to note that this finding is in line with previous research on human scream perception:

Schwartz, J. W., Engelberg, J. W., & Gouzoules, H. (2020). Was that a scream? Listener agreement and major distinguishing acoustic features. *Journal of Nonverbal Behavior*, 44, 233-252.

Engelberg, J. W., Schwartz, J. W., & Gouzoules, H. (2021). The emotional canvas of human screams: Patterns and acoustic cues in the perceptual categorization of a basic call type. *PeerJ*, 9, e10990.

--LI. 745-746: Amplitude is often excluded from similar studies because the amplitude of an audio recording depends not only on the amplitude of the source but also the distance to the microphone. Given that the latter was standardized in these recordings, is there anything preventing the inclusion of amplitude in the analyses?

REVIEWERS' COMMENTS:

Reviewer #1 (Remarks to the Author):

The current manuscript reports on the findings of an acoustic analysis and four behavioural experiments examining the perception and production of volitional non-verbal fear, aggression and pain vocalisations by deaf individuals. The authors conclude that non-verbal vocalisations produced by deaf individuals differ in their acoustic properties from non-verbal vocalisations produced by hearing individuals. Specifically, deaf vocalisations tended to be more harmonic and included less evidence for vocal roughness. Across 4 perceptual experiments, the authors confirm that listeners cannot recognise deaf vocalisations as accurately as they can recognise vocalisations produced by hearing individuals. This effect is most pronounced for aggression vocalisations, and can still be observed for pain vocalisations but is from what I can see much diminished or even absent for fear vocalisations.

Overall the study poses an interesting and well-motivated research question. The study design is clearly explained, analyses appear appropriate and the paper is overall clearly written. However, the paper was written in a very condensed way, such that at least one (Exp2) of the listening experiments is barely analysed/discussed, another is not motivated in the text (Exp4), and almost all analytical details related to the complex acoustic analysis are reported in the supplementary materials. Further, although this cannot be readily changed now, I unfortunately think that the insights we're getting from this paper are somewhat incomplete and believe that including more vocalisations would have made this study much more impactful.

Below I list some additional concerns about the design, presentation, and interpretation of the study.

Thank you for your positive evaluation and very useful comments, which we address in line below. Please also refer to the revised manuscript and supplements in which all revisions are highlighted in blue text. We would like to clarify that the analytic details were in fact provided in the main methods, albeit at the end of the manuscript. The methods section has now been moved before the results.

1a. Why were only aggression, fear, and pain vocalisations used? There are many other types of non-verbal vocalisations the humans produce and I cannot see a satisfactory explanation for this choice of vocalisations in the manuscript. For example, wouldn't it have been useful to also include positively or at least not clearly negatively valenced vocalisations?

This is a valid point that we now explicitly underscore throughout our manuscript.

Our decision to focus on aggression, pain, and fear contexts was motivated by past work on volitional nonverbal vocalisations in typically hearing human participants, most of which has focused on negatively valenced contexts, especially the few studies that included acoustic analyses rather than only perception data (e.g., Raine et al., 2018, 2019; Anikin & Lima, 2018; Kleisner et al., 2022; see also Kamiloglu et

al., 2020 for recent discussion on the subject). Owing to a strongly comparative framework in the evolutionary voice sciences, the bias derives from research on non-human animals where vocal behaviour in agonistic and distress contexts is most intensively studied due to its evolutionary relevance, wherein strong selection pressures have led to salient form-function mappings in these call types (e.g., Fitch et al., 2002; Lingle et al., 2012). Our decision was motivated in part to maximise comparability with past work, both in non-human animals and humans. In addition, time constraints were imposed with our hearing-impaired volunteers, who also needed to complete several hearing tests and surveys during their visit to the lab.

1b. More importantly: Is the conclusion the paper draws from the data likely to generalise beyond these particular vocalisations and can we really glean a full-enough picture from the data at hand? Different vocalisations behave in different ways across the analyses presented in the paper - with fear vocalisations being both in terms of perception and production similar across deaf and hearing producers. Furthermore, the introduction points out that, for example, some studies show that (admittedly spontaneous) laughter vocalisations produced by deaf individuals can be recognised as such. Is it thus not likely that the claim the authors make about auditory feedback being necessary only holds for some vocalisations? I feel like the paper could make this point much clearer.

We were careful to emphasize in our paper that deaf individuals produced atypical vocalisations of aggression and pain, both in acoustic form and perception, whereas fear vocalisations were indeed relatively typical compared to those of healthy controls. This is stated explicitly in the abstract and reiterated in the results, with several paragraphs dedicated to this finding in the discussion. This is indeed a key and interesting result of the study.

We have as suggested further revised the manuscript to make this point even clearer by narrowing general statements (e.g., “Our results indicate that the rare human capacity to voluntarily express *some* emotions through nonverbal vocalizations requires auditory experience to develop typically”). We also now further underscore the limitation of focusing on negative contexts only, and encourage research on a broader range of affective (especially positive) contexts to dig deeper into the potential call-type or context-specificity of auditory experience on vocal production in humans:

“In this study we examined only negatively valenced affective states. With the exception of laughter, researchers have indeed largely focused on negative call types in humans such as screams of fear, roars of aggression, and cries of pain, with far fewer studies on the communication of positive affective states² (but see ⁹³ for review on positive vocalizations). In the comparative voice sciences, this negative bias likely originates from research on non-human animals where vocal behavior in agonistic and distress contexts is most intensively studied due to its obvious evolutionary relevance, with strong selection pressures leading to salient form-function mappings in these call types (^{2,4,28,82} for reviews). More research is thus needed to test for form-function mappings in positively valenced call types in both the typically hearing and hearing-impaired communities.” (Line 890)

2. Exp2: The free text data from Exp2 are potentially quite rich and I feel like there are more impactful ways of describing or reporting these data than is done at the moment. For now only a couple of examples are given of the relative frequency with which certain emotion labels were used across vocalisations and groups, alongside a basic valence assessment of the words. Have the authors, for example, considered plotting the responses as a word cloud to be included as a subpanel in a figure in the main text? I'm not insisting on this being done but I feel like currently the data from this experiment are only presented and discussed very superficially and in a way that is quite removed from the data, such that it makes me wonder why the authors choose this particular task.

Thank you for this excellent suggestion. We now provide a deeper analysis of the data collected during the open-ended emotion identification task, including a new panel in Figure 2 (panel D) in which we show the frequency of occurrence for the top 15 one-word responses given for each emotional context, and for each vocalizer group. We think this figure effectively illustrates that labels attributed to typically hearing vocalizers were far more appropriate than those attributed to deaf vocalizers, particularly in the aggressive context, in which the labels 'fear' and 'pain' were used most often to describe aggressive vocalizations produced by deaf vocalizers.

We have also created a new supplementary figure (Figure S2) on the reviewer's recommendation, presenting word clouds for each context and vocalizer group.

The new text in the Results now reads:

"Figure 2d shows the top 15 words attributed to vocalizations based on their intended emotion and vocalizer group (see also Fig S2 for word clouds and Table S6 for the full list of one-word responses). For typically hearing controls, listeners correctly attributed the labels 'anger', 'pain', and 'fear' to the appropriate vocalizations most of the time, accounting for nearly 300 responses in each emotional context. In contrast, for deaf vocalizers, listeners attributed far fewer appropriate labels to each intended emotion and, in the case of aggressive vocalizations, were more likely to misattribute labels such as 'fear' (193 responses) or 'pain' (133 responses) rather than 'anger' (91 responses) or 'aggression' (7 responses) (Figs 2d, S2). Indeed, the word 'anger' was used three times more often to describe the aggressive vocalizations of controls (293 responses) than those of deaf vocalizers (91 responses). The open-ended emotion labels attributed to deaf vocalizers were also more likely to be positively valenced such as 'joy', which was in the top 5 words attributed to deaf vocalizers across all three emotional contexts, or 'excitement', 'pleased' and 'relief' which were in the top 15 words (Fig 2d)." (Line 602)

3. Exp4: I was interested in why the measure of authenticity was included. It is not mentioned in the introduction, the findings are not discussed at length or tied back to another literature. I'm not sure if the authors would like to tie this scale better into the main narrative or whether they might perhaps prefer to move this scale to the supplementary materials. However, at the moment without further background and context, I did not find the reporting of this scale particularly satisfying as a reader.

Related to this, when reading the prediction for the perceptual experiments, I was unsure where the prediction "Based on our acoustic analyses, we predicted that listeners would judge the vocalizations of deaf men and women as least authentic" came from. Perhaps a couple of lines of explanation behind the rationale of this prediction would be useful.

We have now added additional rationale in the Introduction for testing perceived authenticity, along with additional rationale for all four perception experiments. We have integrated this new text into the final paragraph of the introduction. The new text specific to the authenticity experiment is pasted below:

"In experiment 3, we tested the prediction that listeners would judge the vocalizations of deaf individuals as less authentic than those of hearing controls. Emotion authenticity ratings can index how convincingly a given vocalization conveys an emotion⁵⁷. Generally, the stronger the acoustic form-function mapping, the more authentic a vocalization tends to sound⁵⁸, suggesting that acoustically atypical expressions of aggression, fear, and pain will be judged as relatively inauthentic." (Line 197)

We also expanded on the implications of the results from the authenticity experiment in the Discussion:

"Listeners also judged the vocalizations of deaf men and women as less authentic compared to those of controls, except for women's fear calls. Authenticity ratings can act as a proxy of how genuine or convincing a vocalization sounds to listeners in terms of conveying its intended emotion. Given atypicality in the acoustic forms of aggressive and pain vocalizations produced by deaf vocalizers, it's not surprising that they also sounded less authentic than did those of hearing adults. Authenticity ratings have been shown to positively predict perceptions of affective arousal and person traits such as trustworthiness⁵⁸, broadening the social implications of such attributions." (Line 746)

4. Very minor point for Figure 1B: I don't know if this is intentional or not but the individual data points (i.e. black dots) are not centred on the violin plots. If not intentional, the authors might want to align the dots with the violins.

The overlaid dot plots were indeed intentionally shifted to improve visualization and this has now been noted in the figure caption.

Reviewer #2 (Remarks to the Author):

Based on production (n = 120) and perceptual (n = 444) studies, Pisanski et al. make the case that volitional emotional vocalizations require learning to develop typically. The focus is on nonverbal vocalizations expressing aggression, pain and fear, and the main findings are that (1) deaf adults produce vocalizations that are acoustically atypical, and that such vocalizations (2) are less efficient at communicating the intended emotions, are perceived as less authentic, and are detected as being produced by deaf speakers.

This is a very well-written and well-powered study, the methods are sound, the research question is timely and well-motivated, and the level of detail and rigor of the analyses are impressive. This work also represents a novel and significant contribution to the literature, which is why I recommend publication, even though I have suggestions for improvement that could be addressed in a revision.

Thank you very much for your positive evaluation, your time and your helpful comments, which we address in line below. Please also refer to the revised manuscript in which all edits are highlighted with blue text.

- In the Introduction (lines 91-94), I suggest that the authors avoid the outdated and oversimplified view that the limbic system corresponds to the ‘emotional’ brain. Other regions and networks support emotional processes, at subcortical and cortical levels (for more nuanced accounts, see e.g., Pessoa, 2023, *Neuroscience and Biobehavioral Reviews*; Lindquist et al., 2016, *Cerebral Cortex*).

We agree that this view is overly simplistic, as emotional processing does indeed to implicate multiple areas as the brain and as such, we’ve removed this text.

- The authors’ argument is based primarily on the role of auditory input, and much less emphasis is placed on how deaf individuals also differ in vocal control, experience, and expertise (and on how such differences potentially contribute to the observed effects). It is likely that both lack of auditory input and differences in vocal production skills play a role, and it would be helpful to emphasize the two factors in a more balanced way, or to justify the preferential focus on perceptual aspects. Perhaps it would make sense to speak more of auditory-motor experience throughout the paper, and less of auditory input alone.

We have revised the text throughout the manuscript to be more inclusive in terms of the potential mechanisms that may contribute to vocal atypicality in deaf individuals. We explicitly refer to differences in experience with both auditory input and vocal production in deaf compared to typically hearing individuals (auditory/vocal motor experience).

In our original discussion, we had already discussed this point, however we have now expanded this text to include additional discussion of vocal control and vocal production mechanisms, citing several new works on this topic (new text is underlined below):

“In humans, both auditory input and auditory feedback mechanisms appear to independently contribute to normal speech development⁸⁹. A lack of auditory feedback to monitor vocal production is thought to be an especially important contributor to the impaired vocal control abilities observed during speech production in deaf people, including impaired control over voice fundamental frequency and articulation (⁸² for review). Nevertheless, directly comparing the relative contributions of auditory input and feedback mechanisms, and their potential mediating effects on vocal control, is often not possible in studies of humans with natural causes of deafness, where all of these mechanisms are simultaneously affected...” (Line 857)

“Several scholars have suggested that differences in the emotive expressions of sensory-impaired versus typically developing persons may also arise for reasons other than obstructed auditory input or vocal learning^{52,91}. For instance, infrequent vocal production may affect vocal fold response properties and oral muscle control⁵², including poor coordination of laryngeal muscles and abnormalities in their tension that can directly affect voice pitch and its stability⁸⁸. Infrequent use of the tongue or coordination of other articulators (such as the jaw and lips) can lead to physical changes that may affect articulation, and even lung capacity can differ in deaf versus hearing individuals, affecting control of breathing and thus rate and fluency of vocal signals⁸⁸.” (Line 861).

- Related to the previous point, finding that the effects are larger for vocalisers with less auditory experience is not strong evidence for the role of auditory input, because these individuals are likely to also have less vocal production experience. In fact, I was wondering whether the authors have any measure of speech/voice production experience/proficiency, which they could correlate with their measures of interest. If not, this could be discussed a potential avenue for future work.

While we did quantify the severity of hearing loss in our sample, we did not have any measures of vocal production proficiency. We have noted this as an important avenue for future work that we agree could help to tease apart the relative contributions of auditory experience and vocal production experience in the vocal communication of deaf individuals:

“To tease apart the relative contributions of auditory experience versus vocal production experience in the development of nonverbal vocalizations, researchers will need to measure not only the severity of hearing loss in deaf volunteers, but also their overall experience with vocal production, alongside speech articulation and vocal control tests.” (Line 868).

We have also revised the wording throughout the manuscript to clarify this point, here are just a few examples:

“These results confirm that the more severe a vocalizer’s auditory deficit, the less typical their vocalizations sound. This strongly suggests that acoustic experience from the outside world plays a role in the formation of volitional nonverbal vocalizations by facilitating vocal learning and/or vocal motor control, particularly for vocalizations communicating aggressive intent.”

“This strongly suggests that the effects of deafness on vocal production are “additive”. However, multiple direct and indirect mechanisms may be at play.”

“Because aggressive roars emerge later in human ontogeny than do distress vocalizations or cries, they may require more auditory and/or vocal motor experience to develop typically into adulthood.

“These results implicate vocal learning as a potential key player not only in the acquisition of speech, but also in the acquisition of non-linguistic human

vocalizations, though a potential role of vocal motor experience cannot be excluded.”

- Can the authors explain why they decided to focus on aggression, pain and fear vocalisations, specifically? I was surprised that none of the vocalizations was positive. Issues of generalisability could be discussed in more detail.

This is a good point that was also raised by the first Reviewer and that we have been careful to justify in the manuscript. Our decision to focus on aggression, pain, and fear contexts was motivated by past work on volitional nonverbal vocalisations in normally hearing human participants, much of which has focused on negatively valenced contexts, especially the few studies that included acoustic analysis alongside perception data (e.g., Raine et al., 2018, 2019; Pisanski et al., 2022 for review). This bias toward negative contexts derives from research on non-human animals where vocal behaviour in agonistic and distress contexts is most intensively studied due to its obvious evolutionary relevance (see e.g., Fitch et al., 2002; Lingle et al., 2012). Thus, our decision was motivated in part to maximise comparability with past work, within the time constraints we had to impose with our hearing-impaired volunteers, who also needed to complete several hearing tests and surveys during their visit to the lab.

We now explain this rationale and underscore the limitation of focusing on negative contexts only, and encourage research on a broader range of affective (especially positive) contexts to dig deeper into the potential call-type or context-specificity of vocal learning in humans:

“In this study we examined only negatively valenced affective states. With the exception of laughter, researchers have indeed largely focused on negative call types in humans such as screams of fear, roars of aggression, and cries of pain, with far fewer studies on the communication of positive affective states² (but see ⁹³ for review on positive vocalizations). In the comparative voice sciences, this negative bias likely originates from research on non-human animals where vocal behavior in agonistic and distress contexts is most intensively studied due to its obvious evolutionary relevance, with strong selection pressures leading to salient form-function mappings in these call types (^{2,4,28,82} for reviews). More research is thus needed to test for form-function mappings in positively valenced call types in both the typically hearing and hearing-impaired communities.” (Line 890)

- Using the word ‘authenticity’ when all vocalisations are non-authentic reads like a misnomer. Would expressions like prototypical, typical, or representative be more appropriate?

We have added additional text and predictions to the Introduction to clarify our rationale for testing perceived authenticity, and further discuss the implications in the Discussion. Although authenticity ratings can be used to test whether listeners can discriminate “spontaneous” versus “volitional” vocalisations, authenticity ratings have also been used to examine how ‘convincing’ or genuine a vocalisation sounds and how this can affect other social evaluations of vocalisers (Anikin & Lima 2018; Pinheiro et al., 2021). Our new text reads:

“Emotion authenticity ratings can index how convincingly a given vocalization conveys an emotion⁵⁷. Generally, the stronger the acoustic form-function mapping, the more authentic a vocalization tends to sound⁵⁸, suggesting that acoustically atypical expressions of aggression, fear, and pain will be judged as relatively inauthentic.” (Line 199)

“Listeners also judged the vocalizations of deaf men and women as less authentic compared to those of controls, except for women’s fear calls. Authenticity ratings can act as a proxy of how genuine or convincing a vocalization sounds to listeners in terms of conveying its intended emotion. Given atypicality in the acoustic forms of aggressive and pain vocalizations produced by deaf vocalizers, it’s not surprising that they also sounded less authentic than did those of hearing adults. Authenticity ratings have been shown to positively predict perceptions of affective arousal and person traits such as trustworthiness⁵⁸, broadening the social implications of such attributions.” (Line 746)

- Recognition accuracy: do the findings replicate when accuracy rates are corrected for possible response biases, using for example unbiased hit rates, Hu scores (Wagner, 1993, Journal of Nonverbal Behavior)?

We thank the reviewer for this excellent suggestion. Indeed, our confusion plots had showed evidence of a response bias toward judging vocalizations of deaf vocalizers as fearful. As suggested, we thus computed Hu scores for each individual (see new Table S14) and arcsine-transformed the scores for use in Wilcoxon signed-rank tests which revealed even stronger differences in listeners classification accuracy for all emotions, including aggression, pain and fear, and for both sexes (Table S15). Across the board, listeners were more accurate in judging the intended emotions of hearing compared to deaf vocalizers when response biases were controlled for. We have now added descriptions of these new analyses to the statistical analysis and results sections and added the mean Hu scores alongside the raw hit rates in Figure 1a.

- I would expect recognition accuracy to be higher than 60% for typical vocalisations, considering that accuracy rates are usually high for nonverbal vocalisations (e.g., Laukka & Elenfeldt, 2021, Emotion Review; Sauter et al., 2010, Quarterly Journal Experimental Psychology; Belin et al., 2008, Behavior Research Methods; Lima et al., 2013, Behavior Research Methods) and that there were only three options. Can the authors comment on that?

The above referenced papers that examined emotion recognition in normally hearing individuals used a wider range of affective contexts, including a mix of both negatively and positively valenced call types (e.g., happiness, pleasure, calmness). Here, as noted above, we focused on three ecologically valid contexts that are all negatively valenced. Valence can explain a fair amount of the variance in listeners’ voice-based assessments of emotions as most errors are made for emotions of the same valence (e.g., Sauter et al., 2010), and so we can expect a higher degree of confusion among these emotions, and thus lower accuracy, when assessing only negative emotion types. Nevertheless, accuracy of 60% was still almost double the

chance level (33%) and exceeded 70% for aggression in normally hearing vocalizers.

Sauter, D. A., Eisner, F., Calder, A. J., & Scott, S. K. (2010). Perceptual cues in nonverbal vocal expressions of emotion. *Quarterly journal of experimental psychology*, 63(11), 2251-2272.

- Recordings: If I understood correctly, each vocaliser recorded only one vocalisation per emotion, without time for familiarisation with the task and rehearsal. In my experience, the production of vocalizations on command is perceived as an unusual task, often difficult, and for many speakers it takes time, both for them to understand what is required and to feel comfortable producing the sounds. The potential role of these factors might be even more important for deaf vocalizers – for them the task is likely to be more difficult. Can these confounds explain some of the observed effects?

We acknowledged in our original discussion the potential social stigmas surrounding voice production in deaf individuals that may affect vocal behavior (Line 872).

However, we can be confident that participants, both deaf and hearing, understood the task well. As explained in the Methods, all participants completed the voice recording task in individual, in-lab sessions in which researchers ensured task comprehension before any recordings were made. Deaf participants were provided with both written instructions and watched a pre-recorded video with the same instructions provided in sign-language. In addition, a sign-language interpreter was present throughout the entire experiment for all deaf participants to answer any questions.

Still, we agree that we cannot exclude the possibility that producing volitional vocalisations can be embarrassing for some individuals. To reduce this factor, participants were left alone in the recording room during the actual voice production task. Our personal experience shows that privacy greatly reduces inhibition. We have now noted this as a potential and indeed general limitation of research on volitional vocal behaviour:

“It should be noted that producing emotional vocalizations on demand can feel embarrassing for some people, whether or not they are hearing impaired, and this can potentially affect the degree to which posed vocal displays reflect their spontaneous counterparts. To alleviate this, all vocalizers in our study were left alone during the voice recording task as privacy is known to reduce such inhibitions.” (Line 899)

Reviewer #3 (Remarks to the Author):

In their submitted manuscript, Pisanski, Reby, and Oleszkiewicz report the results of an extensive study of volitional nonverbal vocalizations of aggression, pain, and fear, focusing on differences in the production and perception of these vocalizations as produced by deaf vs. hearing vocalizers. Acoustic analyses clearly showed average differences in the qualities of the vocalizations. In four perceptual experiments,

listeners judged emotion from the deaf-produced vocalizations significantly less accurately, and perceived those vocalizations as significantly less authentic, compared to hearing-produced vocalizations. Listeners were also significantly accurate at detecting whether a given vocalization was produced by a deaf vocalizer. The researchers report the effects of intended emotion and of vocalizer sex. The central conclusion, that voluntary production of typical emotional vocalizations requires vocal learning, is well supported by the data.

This study represents a substantial effort and carries the potential to significantly advance the field of nonverbal vocal communication. I read the manuscript with great interest, and carefully reviewed the researchers' experimental, acoustic, and statistical methods, results, and interpretation. I was impressed with the comprehensiveness of the research, the novel study conceptualization, and the thorough and well-articulated discussion. I have recommendations regarding statistical analysis and some minor revisions to incorporate additional relevant literature, as well as a question about acoustic variables.

Thank you for this very positive feedback and for your constructive comments which in address in line below. Revisions to the manuscript are shown in blue text in the revised version.

Statistical Analyses

--My understanding is that in Experiments 1 and 4, the authors first computed each listener's proportion of accurate responses for each emotion category and vocalizer sex, then used that as the dependent variable in subsequent analyses (LMM). Rather than aggregating the data in this way, I believe it would be more powerful to use a binomial GLMM with individual responses, each coded as accurate or inaccurate, as the dependent variable.

Indeed, the reviewer's interpretation is correct. Our decision to run linear mixed models on the proportion of correct responses as a dependant variable was based namely on the nature of the task in Experiment 1, in which listeners chose between three emotional categories, thus the original response data were not binary.

We have however now run new models as proposed using GLMM (binary logistic regressions) with the listeners responses coded as correct or incorrect (0,1). Other than the binary dependent variable, the models followed the same structure as the LMMs. We have included the results of these GLMMs in Tables S13b (for Experiment 1) and S17b (for Experiment 4), along with all model effects and parameters in the footnotes. The results are identical to those obtained for the LMMs in terms of significance for all main and interaction effects. We have now made note of this in the statistical analysis and results sections of the manuscript.

Minor Comments

--LI. 85-87 "While the forms and functions of many human vocalizations appear homologous to those of other animals, there is one critical exception: humans can effortlessly, voluntarily produce vocal sounds unlike any other terrestrial mammal, including other primates (3,29)." This seems to imply that other species are incapable of volitional production of vocalizations. Such a view is represented in the literature, but does not reflect the consensus. It is clear that the human ability to modulate the voice far exceeds that of all other terrestrial mammals, but there is

some evidence that other species can volitionally use vocalizations. See, e.g.: Schel, A. M., Townsend, S. W., Machanda, Z., Zuberbühler, K., & Slocombe, K. E. (2013). Chimpanzee alarm call production meets key criteria for intentionality. *PLoS One*, 8(10), e76674.

Ghazanfar, A. A., Liao, D. A., & Takahashi, D. Y. (2019). Volition and learning in primate vocal behaviour. *Animal Behaviour*, 151, 239-247.

For a review, see: Schwartz, J. W., Engelberg, J. W., & Gouzoules, H. (2020). Evolving views on cognition in animal vocal communication: contributions from scream research. *Animal Behavior and Cognition*, 7(2), 192-213.

We agree, as noted in the sentence that proceeds the one noted here. However, we have now further revised this text to avoid any confusion. We clarify that vocal modulation is more *advanced* in humans than in most other mammals, most notably other primates, without implying that vocal flexibility is absent in other primates. We have also added the review paper cited by the Reviewer.

“While the forms and functions of many human vocalizations appear homologous to those of other animals, there is one critical exception: humans can voluntarily produce vocal sounds more effortlessly than any other primate species^{3,31}. Although other primates including great apes do show some degree of vocal flexibility (^{32,33} for reviews), their ability to control their vocal output is extremely rudimentary compared to that of humans³.” (Line 86)

We have also added the following text to the Introduction (new text underlined):

“Although deafening and isolation experiments in primates are very rare, the vocal repertoires of squirrel monkeys also remain intact despite early-deafening or isolation⁴⁰. This is consistent with a general lack of evidence for vocal production learning in nonhuman primates, including great apes^{36,41,42}. Notably, however, there is mounting evidence for vocal plasticity and a developmental role of experience in shaping the vocalisations of some non-human primates, such as marmosets^{43,44}, as they age.” (Line 119)

--L. 120: The most recent publication on vocal learning in nonhuman primates cited here is from 1997. I recommend incorporating more recent work on this subject, e.g.: Egnor, S. R., & Hauser, M. D. (2004). A paradox in the evolution of primate vocal learning. *Trends in Neurosciences*, 27(11), 649-654.

Seyfarth, R. M., & Cheney, D. L. (2010). Production, usage, and comprehension in animal vocalizations. *Brain and Language*, 115(1), 92-100.

We have added more recent reviews on vocal production learning including the Seyfarth & Cheney paper noted by the reviewer, and a recent review paper from the 2021 theme issue on “Vocal learning in animals and humans” published in *Philosophical Transactions of the Royal Society B*:

Janik, V. M., & Knörnschild, M. (2021). Vocal production learning in mammals revisited. *Philosophical Transactions of the Royal Society B*, 376(1836), 20200244.

--Discussion: Regarding the finding that listeners perceived higher-pitched and tonal vocalizations as more fearful (including aggressive vocalizations produced by deaf individuals), it might be relevant to note that this finding is in line with previous research on human scream perception:

Schwartz, J. W., Engelberg, J. W., & Gouzoules, H. (2020). Was that a scream? Listener agreement and major distinguishing acoustic features. *Journal of Nonverbal Behavior*, 44, 233-252.

Engelberg, J. W., Schwartz, J. W., & Gouzoules, H. (2021). The emotional canvas of human screams: Patterns and acoustic cues in the perceptual categorization of a basic call type. *PeerJ*, 9, e10990.

We have expanded the text to note that our results indeed do corroborate research on fear perception in human screams, citing the recommended paper (ref 24), which we have now also cited in several other relevant spots in the manuscript, and thank for reviewer for recommending it. The new text (underlined) reads:

“Most notably, deaf adults produced aggressive and pain calls that were unusually high pitched and tonal, with wide formant spacing and a lack of articulation. In turn, listeners often misidentified the intended emotion or valence of these vocalizations when produced by deaf adults, judging a disproportionate portion of these calls as fearful. This corroborates predictions arising from conventional form-function mappings, wherein high-pitched vocalizations are typically perceived as communicating fear or distress across animal species^{5,81}, with pitch also explaining the majority of the variance in fear perception from human screams²⁴” (Line 737)

--LI. 745-746: Amplitude is often excluded from similar studies because the amplitude of an audio recording depends not only on the amplitude of the source but also the distance to the microphone. Given that the latter was standardized in these recordings, is there anything preventing the inclusion of amplitude in the analyses?

Indeed, because we standardized the distance of the vocalizers from the microphone, we were able to include amplitude in the acoustic analysis. As noted in our methods, “Amplitude parameters included mean intensity (meanAMP), max intensity (maxAMP), and intensity variability (intCV, the coefficient of variation of the intensity contour).” And as noted in our results, we did not find substantial differences in amplitude between the deaf and normally hearing groups of vocalizers. In deaf men, we note that “pain vocalizations were approximately 10% lower in amplitude, on average, than were those of male controls (Table S8)”, but we specify that this difference was not significant. To drive the point home, we have now added a line in our discussion reiterating that we did not find significant differences in amplitude between our groups.

30th Jan 24

Dear Dr Pisanski,

Thank you for submitting your manuscript titled "Humans need auditory experience to produce typical volitional nonverbal vocalizations" to Communications Psychology. However, due to certain shortcomings we are concerned that sending the current manuscript out to review could lead to unnecessary delays and quite possibly an undesirable outcome of the review process.

In particular, as we let you know in our last decision letter, we need you to you to comply with our statistics guidelines. We have attached a checklist that we hope will help you to revise the manuscript accordingly. You should ensure they comply with all points, in particular:

- For manuscripts that interpret null results, we require Bayes Factors or equivalence tests to interpret the null results (see attachment for guidance)
- All main analyses should be in the main manuscript, not the Supplement
- All statements or interpretations of your results must be supported by appropriate, fully reported statistics (see attachment for detailed guidance)

We would therefore like to invite you to revise your manuscript to address these concerns before we make a final determination on whether to send your manuscript for external review.

We shall hope to receive your revised version as soon as you are able to complete the suggested revisions. If something similar is published in the interim we will have to consider the impact it has on the novelty of a revised manuscript.

If you anticipate a delay of more than four weeks, please let us know. Should your manuscript be substantially delayed without notifying us in advance and your article is eventually published, the received date may be that of the revised, not the original, version.

If you are not interested in submitting a suitably revised manuscript in the future please let me know immediately so we can close your file. If you have any questions, please contact me.

Please use the link below when you are prepared to resubmit.

[link redacted]

Thank you for your interest in Communications Psychology.

Best regards,
Saloni Krishnan

Saloni Krishnan, PhD
Editorial Board Member
Communications Psychology
orcid.org/0000-0002-6466-141X

REVIEWERS' COMMENTS:

Reviewer #1 (Remarks to the Author):

The current manuscript reports on the findings of an acoustic analysis and four behavioural experiments examining the perception and production of volitional non-verbal fear, aggression and pain vocalisations by deaf individuals. The authors conclude that non-verbal vocalisations produced by deaf individuals differ in their acoustic properties from non-verbal vocalisations produced by hearing individuals. Specifically, deaf vocalisations tended to be more harmonic and included less evidence for vocal roughness. Across 4 perceptual experiments, the authors confirm that listeners cannot recognise deaf vocalisations as accurately as they can recognise vocalisations produced by hearing individuals. This effect is most pronounced for aggression vocalisations, and can still be observed for pain vocalisations but is from what I can see much diminished or even absent for fear vocalisations.

Overall the study poses an interesting and well-motivated research question. The study design is clearly explained, analyses appear appropriate and the paper is overall clearly written. However, the paper was written in a very condensed way, such that at least one (Exp2) of the listening experiments is barely analysed/discussed, another is not motivated in the text (Exp4), and almost all analytical details related to the complex acoustic analysis are reported in the supplementary materials. Further, although this cannot be readily changed now, I unfortunately think that the insights we're getting from this paper are somewhat incomplete and believe that including more vocalisations would have made this study much more impactful.

Below I list some additional concerns about the design, presentation, and interpretation of the study.

Thank you for your positive evaluation and very useful comments, which we address in line below. Please also refer to the revised manuscript and supplements in which all revisions are highlighted in blue text. We would like to clarify that the analytic details were in fact provided in the main methods, albeit at the end of the manuscript. The methods section has now been moved before the results.

1a. Why were only aggression, fear, and pain vocalisations used? There are many other types of non-verbal vocalisations the humans produce and I cannot see a satisfactory explanation for this choice of vocalisations in the manuscript. For example, wouldn't it have been useful to also include positively or at least not clearly negatively valenced vocalisations?

This is a valid point that we now explicitly underscore throughout our manuscript.

Our decision to focus on aggression, pain, and fear contexts was motivated by past work on volitional nonverbal vocalisations in typically hearing human participants, most of which has focused on negatively valenced contexts, especially the few studies that included acoustic analyses rather than only perception data (e.g., Raine et al., 2018, 2019; Anikin & Lima, 2018; Kleisner et al., 2022; see also Kamiloglu et

al., 2020 for recent discussion on the subject). Owing to a strongly comparative framework in the evolutionary voice sciences, the bias derives from research on non-human animals where vocal behaviour in agonistic and distress contexts is most intensively studied due to its evolutionary relevance, wherein strong selection pressures have led to salient form-function mappings in these call types (e.g., Fitch et al., 2002; Lingle et al., 2012). Our decision was motivated in part to maximise comparability with past work, both in non-human animals and humans. In addition, time constraints were imposed with our hearing-impaired volunteers, who also needed to complete several hearing tests and surveys during their visit to the lab.

1b. More importantly: Is the conclusion the paper draws from the data likely to generalise beyond these particular vocalisations and can we really glean a full-enough picture from the data at hand? Different vocalisations behave in different ways across the analyses presented in the paper - with fear vocalisations being both in terms of perception and production similar across deaf and hearing producers. Furthermore, the introduction points out that, for example, some studies show that (admittedly spontaneous) laughter vocalisations produced by deaf individuals can be recognised as such. Is it thus not likely that the claim the authors make about auditory feedback being necessary only holds for some vocalisations? I feel like the paper could make this point much clearer.

We were careful to emphasize in our paper that deaf individuals produced atypical vocalisations of aggression and pain, both in acoustic form and perception, whereas fear vocalisations were indeed relatively typical compared to those of healthy controls. This is stated explicitly in the abstract and reiterated in the results, with several paragraphs dedicated to this finding in the discussion. This is indeed a key and interesting result of the study.

We have as suggested further revised the manuscript to make this point even clearer by narrowing general statements (e.g., “Our results indicate that the rare human capacity to voluntarily express *some* emotions through nonverbal vocalizations requires auditory experience to develop typically”). We also now further underscore the limitation of focusing on negative contexts only, and encourage research on a broader range of affective (especially positive) contexts to dig deeper into the potential call-type or context-specificity of auditory experience on vocal production in humans:

“In this study we examined only negatively valenced affective states. With the exception of laughter, researchers have indeed largely focused on negative call types in humans such as screams of fear, roars of aggression, and cries of pain, with far fewer studies on the communication of positive affective states² (but see ⁹⁴ for review on positive vocalizations). In the comparative voice sciences, this negative bias likely originates from research on non-human animals where vocal behavior in agonistic and distress contexts is most intensively studied due to its obvious evolutionary relevance, with strong selection pressures leading to salient form-function mappings in these call types (^{2,4,28,83} for reviews). More research is thus needed to test for form-function mappings in positively valenced call types in both the typically hearing and hearing-impaired communities.” (Line 919)

2. Exp2: The free text data from Exp2 are potentially quite rich and I feel like there are more impactful ways of describing or reporting these data than is done at the moment. For now only a couple of examples are given of the relative frequency with which certain emotion labels were used across vocalisations and groups, alongside a basic valence assessment of the words. Have the authors, for example, considered plotting the responses as a word cloud to be included as a subpanel in a figure in the main text? I'm not insisting on this being done but I feel like currently the data from this experiment are only presented and discussed very superficially and in a way that is quite removed from the data, such that it makes me wonder why the authors choose this particular task.

Thank you for this excellent suggestion. We now provide a deeper analysis of the data collected during the open-ended emotion identification task, including a new panel in Figure 2 (panel D) in which we show the frequency of occurrence for the top 15 one-word responses given for each emotional context, and for each vocalizer group. We think this figure effectively illustrates that labels attributed to typically hearing vocalizers were far more appropriate than those attributed to deaf vocalizers, particularly in the aggressive context, in which the labels 'fear' and 'pain' were used most often to describe aggressive vocalizations produced by deaf vocalizers.

We have also created a new supplementary figure (Figure S2) on the reviewer's recommendation, presenting word clouds for each context and vocalizer group.

The new text in the Results now reads:

"Figure 2d shows the top 15 words attributed to vocalizations based on their intended emotion and vocalizer group (see also Fig S2 for word clouds and Table S6 for the full list of one-word responses). For typically hearing controls, listeners correctly attributed the labels 'anger', 'pain', and 'fear' to the appropriate vocalizations most of the time, accounting for nearly 300 responses in each emotional context. In contrast, for deaf vocalizers, listeners attributed far fewer appropriate labels to each intended emotion. In the case of aggressive vocalizations, they were much more likely to misattribute labels such as 'fear' (193 responses) or 'pain' (133 responses) rather than 'anger' (91 responses) or 'aggression' (7 responses) to deaf vocalizers (Figs 2d, S2). Indeed, the word 'anger' was used three times more often to describe the aggressive vocalizations of controls (293 responses) than those of deaf vocalizers (91 responses).

The open-ended emotion labels attributed to deaf vocalizers were also more likely to be positively valenced such as 'joy', which was in the top 5 words attributed to deaf vocalizers across all three emotional contexts, or 'excitement', 'pleased' and 'relief' which were in the top 15 words (Fig 2d)." (Line 621)

3. Exp4: I was interested in why the measure of authenticity was included. It is not mentioned in the introduction, the findings are not discussed at length or tied back to another literature. I'm not sure if the authors would like to tie this scale better into the main narrative or whether they might perhaps prefer to move this scale to the supplementary materials. However, at the moment without further background and context, I did not find the reporting of this scale particularly satisfying as a reader.

Related to this, when reading the prediction for the perceptual experiments, I was unsure where the prediction "Based on our acoustic analyses, we predicted that listeners would judge the vocalizations of deaf men and women as least authentic" came from. Perhaps a couple of lines of explanation behind the rationale of this prediction would be useful.

We have now added additional rationale in the Introduction for testing perceived authenticity, along with additional rationale for all four perception experiments. We have integrated this new text into the final paragraph of the introduction. The new text specific to the authenticity experiment is pasted below:

"Emotion authenticity ratings can index how convincingly a given vocalization conveys an emotion⁵⁷. Generally, the stronger the acoustic form-function mapping, the more authentic a vocalization tends to sound⁵⁸, suggesting that acoustically atypical expressions of aggression, fear, and pain will be judged as relatively inauthentic." (Line 199)

We also expanded on the implications of the results from the authenticity experiment in the Discussion:

"Listeners also judged the vocalizations of deaf men and women as less authentic compared to those of controls, except for women's fear calls. Authenticity ratings can act as a proxy of how genuine or convincing a vocalization sounds to listeners in terms of conveying its intended emotion. Given atypicality in the acoustic forms of aggressive and pain vocalizations produced by deaf vocalizers, it's not surprising that they also sounded less authentic than did those of hearing adults. Authenticity ratings have been shown to positively predict perceptions of affective arousal and person traits such as trustworthiness⁵⁸, broadening the social implications of such attributions." (Line 775)

4. Very minor point for Figure 1B: I don't know if this is intentional or not but the individual data points (i.e. black dots) are not centred on the violin plots. If not intentional, the authors might want to align the dots with the violins.

The overlaid dot plots were indeed intentionally shifted to improve visualization and this has now been noted in the figure caption.

Reviewer #2 (Remarks to the Author):

Based on production (n = 120) and perceptual (n = 444) studies, Pisanski et al. make the case that volitional emotional vocalizations require learning to develop typically. The focus is on nonverbal vocalizations expressing aggression, pain and fear, and the main findings are that (1) deaf adults produce vocalizations that are acoustically atypical, and that such vocalizations (2) are less efficient at communicating the intended emotions, are perceived as less authentic, and are detected as being produced by deaf speakers.

This is a very well-written and well-powered study, the methods are sound, the research question is timely and well-motivated, and the level of detail and rigor of the analyses are impressive. This work also represents a novel and significant

contribution to the literature, which is why I recommend publication, even though I have suggestions for improvement that could be addressed in a revision.

Thank you very much for your positive evaluation, your time and your helpful comments, which we address in line below. Please also refer to the revised manuscript in which all edits are highlighted with blue text.

- In the Introduction (lines 91-94), I suggest that the authors avoid the outdated and oversimplified view that the limbic system corresponds to the 'emotional' brain. Other regions and networks support emotional processes, at subcortical and cortical levels (for more nuanced accounts, see e.g., Pessoa, 2023, *Neuroscience and Biobehavioral Reviews*; Lindquist et al., 2016, *Cerebral Cortex*).

We agree that this view is overly simplistic, as emotional processing does indeed to implicate multiple areas as the brain and as such, we've removed this text.

- The authors' argument is based primarily on the role of auditory input, and much less emphasis is placed on how deaf individuals also differ in vocal control, experience, and expertise (and on how such differences potentially contribute to the observed effects). It is likely that both lack of auditory input and differences in vocal production skills play a role, and it would be helpful to emphasize the two factors in a more balanced way, or to justify the preferential focus on perceptual aspects. Perhaps it would make sense to speak more of auditory-motor experience throughout the paper, and less of auditory input alone.

We have revised the text throughout the manuscript to be more inclusive in terms of the potential mechanisms that may contribute to vocal atypicality in deaf individuals. We explicitly refer to differences in experience with both auditory input and vocal production in deaf compared to typically hearing individuals (auditory/vocal motor experience).

In our original discussion, we had already discussed this point, however we have now expanded this text to include additional discussion of vocal control and vocal production mechanisms, citing several new works on this topic (new text is underlined below):

"In humans, both auditory input and auditory feedback mechanisms appear to independently contribute to normal speech development⁹¹. A lack of auditory feedback to monitor vocal production is thought to be an especially important contributor to the impaired vocal control abilities observed during speech production in deaf people, including impaired control over voice fundamental frequency and articulation (⁸⁴ for review). Nevertheless, directly comparing the relative contributions of auditory input and feedback mechanisms, and their potential mediating effects on vocal control, is often not possible in studies of humans with natural causes of deafness, where all of these mechanisms are simultaneously affected..." (Line 882)

"Several scholars have suggested that differences in the emotive expressions of sensory-impaired versus typically developing persons may also arise for reasons other than obstructed auditory input or vocal learning^{52,92}. For

instance, infrequent vocal production may affect vocal fold response properties and oral muscle control⁵², including poor coordination of laryngeal muscles and abnormalities in their tension that can directly affect voice pitch and its stability⁸⁹. Infrequent use of the tongue or coordination of other articulators (such as the jaw and lips) can lead to physical changes that may affect articulation, and even lung capacity can differ in deaf versus hearing individuals, affecting control of breathing and thus rate and fluency of vocal signals⁹³.” (Line 890).

- Related to the previous point, finding that the effects are larger for vocalisers with less auditory experience is not strong evidence for the role of auditory input, because these individuals are likely to also have less vocal production experience. In fact, I was wondering whether the authors have any measure of speech/voice production experience/proficiency, which they could correlate with their measures of interest. If not, this could be discussed a potential avenue for future work.

While we did quantify the severity of hearing loss in our sample, we did not have any measures of vocal production proficiency. We have noted this as an important avenue for future work that we agree could help to tease apart the relative contributions of auditory experience and vocal production experience in the vocal communication of deaf individuals:

“To tease apart the relative contributions of auditory experience versus vocal production experience in the development of nonverbal vocalizations, researchers will need to measure not only the severity of hearing loss in deaf volunteers, but also their overall experience with vocal production, alongside speech articulation and vocal control tests.” (Line 897).

We have also revised the wording throughout the manuscript to clarify this point, here are just a few examples:

“These results confirm that the more severe a vocalizer’s auditory deficit, the less typical their vocalizations sound. This strongly suggests that acoustic experience from the outside world plays a role in the formation of volitional nonverbal vocalizations by facilitating vocal learning and/or vocal motor control, particularly for vocalizations communicating aggressive intent.”

“This strongly suggests that the effects of deafness on vocal production are “additive”. However, multiple direct and indirect mechanisms may be at play.”

“Because aggressive roars emerge later in human ontogeny than do distress vocalizations or cries, they may require more auditory and/or vocal motor experience to develop typically into adulthood.

“These results implicate vocal learning as a potential key player not only in the acquisition of speech, but also in the acquisition of non-linguistic human vocalizations, though a potential role of vocal motor experience cannot be excluded.”

- Can the authors explain why they decided to focus on aggression, pain and fear

vocalisations, specifically? I was surprised that none of the vocalizations was positive. Issues of generalisability could be discussed in more detail.

This is a good point that was also raised by the first Reviewer and that we have been careful to justify in the manuscript. Our decision to focus on aggression, pain, and fear contexts was motivated by past work on volitional nonverbal vocalisations in normally hearing human participants, much of which has focused on negatively valenced contexts, especially the few studies that included acoustic analysis alongside perception data (e.g., Raine et al., 2018, 2019; Pisanski et al., 2022 for review). This bias toward negative contexts derives from research on non-human animals where vocal behaviour in agonistic and distress contexts is most intensively studied due to its obvious evolutionary relevance (see e.g., Fitch et al., 2002; Lingle et al., 2012). Thus, our decision was motivated in part to maximise comparability with past work, within the time constraints we had to impose with our hearing-impaired volunteers, who also needed to complete several hearing tests and surveys during their visit to the lab.

We now explain this rationale and underscore the limitation of focusing on negative contexts only, and encourage research on a broader range of affective (especially positive) contexts to dig deeper into the potential call-type or context-specificity of vocal learning in humans:

“In this study we examined only negatively valenced affective states. With the exception of laughter, researchers have indeed largely focused on negative call types in humans such as screams of fear, roars of aggression, and cries of pain, with far fewer studies on the communication of positive affective states² (but see ⁹⁴ for review on positive vocalizations). In the comparative voice sciences, this negative bias likely originates from research on non-human animals where vocal behavior in agonistic and distress contexts is most intensively studied due to its obvious evolutionary relevance, with strong selection pressures leading to salient form-function mappings in these call types (^{2,4,28,83} for reviews). More research is thus needed to test for form-function mappings in positively valenced call types in both the typically hearing and hearing-impaired communities.” (Line 919)

- Using the word ‘authenticity’ when all vocalisations are non-authentic reads like a misnomer. Would expressions like prototypical, typical, or representative be more appropriate?

We have added additional text and predictions to the Introduction to clarify our rationale for testing perceived authenticity, and further discuss the implications in the Discussion. Although authenticity ratings can be used to test whether listeners can discriminate “spontaneous” versus “volitional” vocalisations, authenticity ratings have also been used to examine how ‘convincing’ or genuine a vocalisation sounds and how this can affect other social evaluations of vocalisers (Anikin & Lima 2018; Pinheiro et al., 2021). Our new text reads:

“Emotion authenticity ratings can index how convincingly a given vocalization conveys an emotion⁵⁷. Generally, the stronger the acoustic form-function mapping, the more authentic a vocalization tends to sound⁵⁸, suggesting that

acoustically atypical expressions of aggression, fear, and pain will be judged as relatively inauthentic.” (Line 199)

“Listeners also judged the vocalizations of deaf men and women as less authentic compared to those of controls, except for women’s fear calls. Authenticity ratings can act as a proxy of how genuine or convincing a vocalization sounds to listeners in terms of conveying its intended emotion. Given atypicality in the acoustic forms of aggressive and pain vocalizations produced by deaf vocalizers, it’s not surprising that they also sounded less authentic than did those of hearing adults. Authenticity ratings have been shown to positively predict perceptions of affective arousal and person traits such as trustworthiness⁵⁸, broadening the social implications of such attributions.” (Line 775)

- Recognition accuracy: do the findings replicate when accuracy rates are corrected for possible response biases, using for example unbiased hit rates, Hu scores (Wagner, 1993, Journal of Nonverbal Behavior)?

We thank the reviewer for this excellent suggestion. Indeed, our confusion plots had showed evidence of a response bias toward judging vocalizations of deaf vocalizers as fearful. As suggested, we thus computed Hu scores for each individual (see new Table S14) and arcsine-transformed the scores for use in Wilcoxon signed-rank tests which revealed even stronger differences in listeners classification accuracy for all emotions, including aggression, pain and fear, and for both sexes (Table S15). Across the board, listeners were more accurate in judging the intended emotions of hearing compared to deaf vocalizers when response biases were controlled for. We have now added descriptions of these new analyses to the statistical analysis and results sections and added the mean Hu scores alongside the raw hit rates in Figure 1a.

- I would expect recognition accuracy to be higher than 60% for typical vocalisations, considering that accuracy rates are usually high for nonverbal vocalisations (e.g., Laukka & Elenfeldt, 2021, Emotion Review; Sauter et al., 2010, Quarterly Journal Experimental Psychology; Belin et al., 2008, Behavior Research Methods; Lima et al., 2013, Behavior Research Methods) and that there were only three options. Can the authors comment on that?

The above referenced papers that examined emotion recognition in normally hearing individuals used a wider range of affective contexts, including a mix of both negatively and positively valenced call types (e.g., happiness, pleasure, calmness). Here, as noted above, we focused on three ecologically valid contexts that are all negatively valenced. Valence can explain a fair amount of the variance in listeners’ voice-based assessments of emotions as most errors are made for emotions of the same valence (e.g., Sauter et al., 2010), and so we can expect a higher degree of confusion among these emotions, and thus lower accuracy, when assessing only negative emotion types. Nevertheless, accuracy of 60% was still almost double the chance level (33%) and exceeded 70% for aggression in normally hearing vocalizers.

Sauter, D. A., Eisner, F., Calder, A. J., & Scott, S. K. (2010). Perceptual cues in nonverbal vocal expressions of emotion. *Quarterly journal of experimental psychology*, 63(11), 2251-2272.

- Recordings: If I understood correctly, each vocaliser recorded only one vocalisation per emotion, without time for familiarisation with the task and rehearsal. In my experience, the production of vocalizations on command is perceived as an unusual task, often difficult, and for many speakers it takes time, both for them to understand what is required and to feel comfortable producing the sounds. The potential role of these factors might be even more important for deaf vocalizers – for them the task is likely to be more difficult. Can these confounds explain some of the observed effects?

We acknowledged in our original discussion the potential social stigmas surrounding voice production in deaf individuals that may affect vocal behavior (Line 872).

However, we can be confident that participants, both deaf and hearing, understood the task well. As explained in the Methods, all participants completed the voice recording task in individual, in-lab sessions in which researchers ensured task comprehension before any recordings were made. Deaf participants were provided with both written instructions and watched a pre-recorded video with the same instructions provided in sign-language. In addition, a sign-language interpreter was present throughout the entire experiment for all deaf participants to answer any questions.

Still, we agree that we cannot exclude the possibility that producing volitional vocalisations can be embarrassing for some individuals. To reduce this factor, participants were left alone in the recording room during the actual voice production task. Our personal experience shows that privacy greatly reduces inhibition. We have now noted this as a potential and indeed general limitation of research on volitional vocal behaviour:

“It should be noted that producing emotional vocalizations on demand can feel embarrassing for some people, whether or not they are hearing impaired, and this can potentially affect the degree to which posed vocal displays reflect their spontaneous counterparts. To alleviate this, all vocalizers in our study were left alone during the voice recording task as privacy is known to reduce such inhibitions.” (Line 928)

Reviewer #3 (Remarks to the Author):

In their submitted manuscript, Pisanski, Reby, and Oleszkiewicz report the results of an extensive study of volitional nonverbal vocalizations of aggression, pain, and fear, focusing on differences in the production and perception of these vocalizations as produced by deaf vs. hearing vocalizers. Acoustic analyses clearly showed average differences in the qualities of the vocalizations. In four perceptual experiments, listeners judged emotion from the deaf-produced vocalizations significantly less accurately, and perceived those vocalizations as significantly less authentic, compared to hearing-produced vocalizations. Listeners were also significantly

accurate at detecting whether a given vocalization was produced by a deaf vocalizer. The researchers report the effects of intended emotion and of vocalizer sex. The central conclusion, that voluntary production of typical emotional vocalizations requires vocal learning, is well supported by the data.

This study represents a substantial effort and carries the potential to significantly advance the field of nonverbal vocal communication. I read the manuscript with great interest, and carefully reviewed the researchers' experimental, acoustic, and statistical methods, results, and interpretation. I was impressed with the comprehensiveness of the research, the novel study conceptualization, and the thorough and well-articulated discussion. I have recommendations regarding statistical analysis and some minor revisions to incorporate additional relevant literature, as well as a question about acoustic variables.

Thank you for this very positive feedback and for your constructive comments which in address in line below. Revisions to the manuscript are shown in blue text in the revised version.

Statistical Analyses

--My understanding is that in Experiments 1 and 4, the authors first computed each listener's proportion of accurate responses for each emotion category and vocalizer sex, then used that as the dependent variable in subsequent analyses (LMM). Rather than aggregating the data in this way, I believe it would be more powerful to use a binomial GLMM with individual responses, each coded as accurate or inaccurate, as the dependent variable.

Indeed, the reviewer's interpretation is correct. Our decision to run linear mixed models on the proportion of correct responses as a dependant variable was based namely on the nature of the task in Experiment 1, in which listeners chose between three emotional categories, thus the original response data were not binary.

We have however now run new models as proposed using GLMM (binary logistic regressions) with the listeners responses coded as correct or incorrect (0,1). Other than the binary dependent variable, the models followed the same structure as the LMMs. We have included the results of these GLMMs in Tables S13b (for Experiment 1) and S17b (for Experiment 4), along with all model effects and parameters in the footnotes. The results are identical to those obtained for the LMMs in terms of significance for all main and interaction effects. We have now made note of this in the statistical analysis and results sections of the manuscript.

Minor Comments

--LI. 85-87 "While the forms and functions of many human vocalizations appear homologous to those of other animals, there is one critical exception: humans can effortlessly, voluntarily produce vocal sounds unlike any other terrestrial mammal, including other primates (3,29)." This seems to imply that other species are incapable of volitional production of vocalizations. Such a view is represented in the literature, but does not reflect the consensus. It is clear that the human ability to modulate the voice far exceeds that of all other terrestrial mammals, but there is some evidence that other species can volitionally use vocalizations. See, e.g.: Schel, A. M., Townsend, S. W., Machanda, Z., Zuberbühler, K., & Slocombe, K. E. (2013). Chimpanzee alarm call production meets key criteria for intentionality. PLoS

One, 8(10), e76674.

Ghazanfar, A. A., Liao, D. A., & Takahashi, D. Y. (2019). Volition and learning in primate vocal behaviour. *Animal Behaviour*, 151, 239-247.

For a review, see: Schwartz, J. W., Engelberg, J. W., & Gouzoules, H. (2020). Evolving views on cognition in animal vocal communication: contributions from scream research. *Animal Behavior and Cognition*, 7(2), 192-213.

We agree, as noted in the sentence that proceeds the one noted here. However, we have now further revised this text to avoid any confusion. We clarify that vocal modulation is more *advanced* in humans than in most other mammals, most notably other primates, without implying that vocal flexibility is absent in other primates. We have also added the review paper cited by the Reviewer.

“While the forms and functions of many human vocalizations appear homologous to those of other animals, there is one critical exception: humans can voluntarily produce vocal sounds more effortlessly than any other primate species^{3,31}. Although other primates including great apes do show some degree of vocal flexibility (^{32,33} for reviews), their ability to control their vocal output is extremely rudimental compared to that of humans³.” (Line 86)

We have also added the following text to the Introduction (new text underlined):

“Although deafening and isolation experiments in primates are very rare, the vocal repertoires of squirrel monkeys also remain intact despite early-deafening or isolation⁴⁰. This is consistent with a general lack of evidence for vocal production learning in nonhuman primates, including great apes^{36,41,42}. Notably, however, there is mounting evidence for vocal plasticity and a developmental role of experience in shaping the vocalisations of some non-human primates, such as marmosets^{43,44}, as they age.” (Line 119)

--L. 120: The most recent publication on vocal learning in nonhuman primates cited here is from 1997. I recommend incorporating more recent work on this subject, e.g.: Egnor, S. R., & Hauser, M. D. (2004). A paradox in the evolution of primate vocal learning. *Trends in Neurosciences*, 27(11), 649-654.

Seyfarth, R. M., & Cheney, D. L. (2010). Production, usage, and comprehension in animal vocalizations. *Brain and Language*, 115(1), 92-100.

We have added more recent reviews on vocal production learning including the Seyfarth & Cheney paper noted by the reviewer, and a recent review paper from the 2021 theme issue on “Vocal learning in animals and humans’ published in *Philosophical Transactions of the Royal Society B*:

Janik, V. M., & Knörnschild, M. (2021). Vocal production learning in mammals revisited. *Philosophical Transactions of the Royal Society B*, 376(1836), 20200244.

--Discussion: Regarding the finding that listeners perceived higher-pitched and tonal vocalizations as more fearful (including aggressive vocalizations produced by deaf individuals), it might be relevant to note that this finding is in line with previous

research on human scream perception:

Schwartz, J. W., Engelberg, J. W., & Gouzoules, H. (2020). Was that a scream? Listener agreement and major distinguishing acoustic features. *Journal of Nonverbal Behavior*, 44, 233-252.

Engelberg, J. W., Schwartz, J. W., & Gouzoules, H. (2021). The emotional canvas of human screams: Patterns and acoustic cues in the perceptual categorization of a basic call type. *PeerJ*, 9, e10990.

We have expanded the text to note that our results indeed do corroborate research on fear perception in human screams, citing the recommended paper (ref 24), which we have now also cited in several other relevant spots in the manuscript, and thank for reviewer for recommending it. The new text (underlined) reads:

“Most notably, deaf adults produced aggressive and pain calls that were unusually high pitched and tonal, with wide formant spacing and a lack of articulation. In turn, listeners often misidentified the intended emotion or valence of these vocalizations when produced by deaf adults, judging a disproportionate portion of these calls as fearful. This corroborates predictions arising from conventional form-function mappings, wherein high-pitched vocalizations are typically perceived as communicating fear or distress across animal species^{5,83}, with pitch also explaining the majority of the variance in fear perception from human screams²⁴” (Line 766)

--LI. 745-746: Amplitude is often excluded from similar studies because the amplitude of an audio recording depends not only on the amplitude of the source but also the distance to the microphone. Given that the latter was standardized in these recordings, is there anything preventing the inclusion of amplitude in the analyses?

Indeed, because we standardized the distance of the vocalizers from the microphone, we were able to include amplitude in the acoustic analysis. As noted in our methods, “Amplitude parameters included mean intensity (meanAMP), max intensity (maxAMP), and intensity variability (intCV, the coefficient of variation of the intensity contour).” And as noted in our results, we did not find substantial differences in amplitude between the deaf and normally hearing groups of vocalizers. In deaf men, we note that “pain vocalizations were approximately 10% lower in amplitude, on average, than were those of male controls (Table S8)”, but we specify that this difference was not significant. To drive the point home, we have now added a line in our discussion reiterating that we did not find significant differences in amplitude between our groups.